# QUANTUM GENERATOR KERNELS

## ABSTRACT

Quantum kernel methods offer significant theoretical benefits by rendering classically inseparable features separable in quantum space. Yet, the practical application of *Quantum Machine Learning* (QML), currently constrained by the limitations of *Noisy Intermediate-Scale Quantum* (NISQ) hardware, necessitates effective strategies to compress and embed large-scale real-world data like images into the constrained capacities of existing quantum devices or simulators. To this end, we propose *Quantum Generator Kernels* (QGKs), a generator-based approach to quantum kernels, comprising a set of *Variational Generator Groups* (VGGs) that merge universal generators into a parameterizable operator, ensuring scalable coverage of the available quantum space. Thereby, we address shortcomings of current leading strategies employing hybrid architectures, which might prevent exploiting quantum computing's full potential due to fixed intermediate embedding processes. To optimize the kernel alignment to the target domain, we train a weight vector to parameterize the projection of the VGGs in the current data context. Our empirical results demonstrate superior projection and classification capabilities of the QGK compared to state-of-the-art quantum and classical kernel approaches and show its potential to serve as a versatile framework for various QML applications.

## 1 INTRODUCTION

Quantum computing offers fundamentally new paradigms for machine learning by exploiting quantum properties such as superposition and entanglement (Preskill, 2018; Biamonte et al., 2017). Among these, quantum kernel methods have shown promise for enhancing data separability via expressive feature maps that operate in high-dimensional Hilbert spaces, allowing them to capture structures that classical kernels cannot efficiently represent (Schuld & Killoran, 2019). Despite these theoretical promises, the practical application of QML is currently limited by the capabilities of NISQ devices, which are still in their developmental stages (Preskill, 2018). Still limited in qubit capacities and subject to errors, embedding large-scale data into quantum devices is a significant hurdle that must be overcome to exploit the full potential of QML. Hybrid QML architectures bridge this gap, using classical pre-processing to embed data into quantum systems (Cerezo et al., 2021). However, recent studies also highlight key limitations: many current approaches rely on fixed embeddings that do not scale well with high-dimensional inputs and are susceptible to barren plateaus during training (McClean et al., 2018). Addressing this requires scalable, flexible, and learnable quantum embeddings that can exploit these properties while remaining parameter-efficient and robust to noisy hardware.

In this work, we propose a novel kernel architecture grounded in Lie algebraic generators, aggregated into parameterizable groups that project input data directly into quantum space. Our approach can be broken down into three steps: Construct a set of generators and merge them into *Variational Generator Groups* (VGGs). We refer to a set of these groups as the *Quantum Generator Kernel* (QGK), which is executed by its set of operators. To improve alignment between data and the resulting kernel, we introduce a linear feature extractor that is pre-trained to project high-dimensional input into a compressed generator space. Unlike previous methods that rely on static gate-based embeddings, our QGK architecture employs Hamiltonian-driven unitaries with learnable generator weights, enabling expressive and scalable data encoding. By projecting high-dimensional data into a compact generator-weighted space, QGKs achieve high parameter efficiency per qubit and flexible embedding capacity, while effectively leveraging the expressive power of the full Hilbert space. We validate the QGK both analytically and empirically: theoretical analyses confirm its expressivity and

scalability, while experimental results demonstrate superior classification accuracy and robustness to noise across synthetic and real-world benchmarks. We summarize our contributions as follows:

- We introduce *Variational Generator Groups* (VGGs), a novel embedding framework that systematically aggregates Lie-algebraic generators into parameterized Hermitian operators. This construction provides a principled and expressive foundation for data-dependent quantum state preparation and evolution and a scalable alternative to fixed, gate-based encoding.

- Building on the VGG framework, we propose the *Quantum Generator Kernel* (QGK), a generator-driven quantum kernel architecture that employs Hamiltonian evolution with data-conditioned generator weights. This yields compact learnable feature maps with favorable parameter–qubit ratios and high representational capacity.

- We provide an empirical analysis of the VGGs' theoretical kernel properties, characterizing their entanglement capability, expressivity, and parameter scalability. We further examine the computational complexity, demonstrating a classically efficient strategy for handling high-dimensional inputs while remaining fully compatible with future fault-tolerant quantum execution.

- We empirically evaluate the QGK on a diverse range of binary and multi-class classification benchmarks, including `MNIST` and `CIFAR10`. Our results show that the QGK consistently outperforms state-of-the-art classical and quantum kernels, maintaining robustness under realistic noise models of current hardware.

## 2 BACKGROUND

**Kernel methods**  Kernel methods describe a map into a high-dimensional feature space $\psi$, used to project a $d$-dimensional data point $x \in \mathbb{R}^d$, into a space, where the distribution of data may then be suitable for linear separation. The kernel function $k(x_i, x_j) = \langle \psi(x_i), \psi(x_j) \rangle$ can be understood as the pairwise distance on that feature space. As the feature map for all points may be computationally expensive, it can be important to reduce the number of calls to that function. The *kernel trick* allows for calculating pairwise distances in the feature space without explicitly calculating the feature function. This is possible, for example, if the kernel is a positive definite map, independent of its dimension (Hofmann et al., 2008). The pairwise distances $k(x_i, x_j)$ can be used in a *support vector machine* (SVM) to solve a classification problem. SVMs turn the classification problem into a quadratic optimization problem, defined on pairwise distances using the kernel trick:

$$\arg \min_{\boldsymbol{\alpha}} \frac{1}{2} \sum_{\substack{i=1 \\ j=1}}^{n} \boldsymbol{\alpha}_i \boldsymbol{\alpha}_j y_i y_j k(x_i, x_j) - \sum_{i=1}^{n} \boldsymbol{\alpha}_i \tag{1}$$

where $\boldsymbol{\alpha}$ are the parameters of the support vector machine and $y$ are the labels of $n$ datapoints. The quality of a feature map can be measured and adjusted to provide a better data embedding structure using kernel target alignment (KTA) (Cristianini et al., 2001). KTA quantifies the similarity between a computed kernel matrix and an ideal target kernel, e.g., derived from the class labels.

**Quantum computation**  Quantum systems are described by quantum states, represented as complex vectors encoding observables like spin or position. A single qubit can be written as $|q\rangle = \boldsymbol{\alpha} |0\rangle + \beta |1\rangle$, where $\boldsymbol{\alpha}, \beta \in \mathbb{C}$. Quantum operations on these states are unitary transformations $\boldsymbol{U}$ such that $|\Psi'\rangle = \boldsymbol{U} |\Psi\rangle$. In practice, these are decomposed into elementary one- and two-qubit gates and executed on quantum hardware. Measurement collapses the quantum state into classical outcomes Nielsen & Chuang (2010). Mathematically, such operations form the special unitary group $SU(N)$ for $N = 2^\eta$ qubits. As a Lie group, $SU(N)$ has a corresponding Lie algebra $\mathfrak{su}(N)$, a real vector space of skew-Hermitian, trace-zero matrices Hall (2013). The elements of the Lie algebra, known as generators, define the fundamental directions in which unitary quantum operations can be constructed. Since the Lie algebra forms a linear vector space, multiple generators can be combined additively to form complex Hermitian operators. These operators are then mapped to unitary matrices via the exponential map, $\exp \colon \mathfrak{su}(N) \to SU(N)$. In quantum computing, a natural basis for this algebra is the Pauli basis, consisting of tensor products of $\sigma_x, \sigma_y, \sigma_z, \mathbb{1}$: These form the building blocks for quantum circuits and support gradient-based learning due to their manifold structure.

**Quantum Kernel Methods**  A central difference in a system of $\eta$ qubits compared to classical bits is that the dimension of the mathematical space of $\eta$ qubits scales exponentially in $\eta$. Formally, the state of a system of qubits is a ray in a *Hilbert space*, which is generally expressed by $\mathbb{C}^{2^\eta}$. This exponential scaling motivates the quantum kernel method, in which the Hilbert space is used as the feature space, analogous to conventional kernel methods (Mengoni & Di Pierro, 2019). As shown in (Schuld et al., 2021a), a large class of supervised quantum models are kernel methods. A deeper analysis of the mathematical structure of data embedding in quantum circuits shows that combining data uploading with parameterized quantum gates allows for arbitrary function approximation in the form of a Fourier series (Schuld et al., 2021b).

## 3 VARIATIONAL GENERATOR GROUPS

Information can be encoded into a quantum system either by initializing a quantum state with free parameters or by applying a parameterized operator to a fixed initial state. Given the limited qubit counts in current NISQ hardware, we aim to maximize parameter density per qubit. A common quantum encoding approach, known as amplitude encoding, embeds classical data directly into a state vector within a $2^\eta$-dimensional Hilbert space, where each entry represents a complex amplitude (Biamonte et al., 2017). Accounting for both the real and imaginary components as well as the normalization constraint, this allows for a total of $2^{\eta+1} - 1$ free parameters. An alternative and more hardware-friendly variant is rotational (angle) encoding, where classical values are mapped to the rotation angles of single-qubit gates (e.g., $R_x(\boldsymbol{\theta})$), applied to each qubit independently. While this method is easier to implement and preserves data locality, it significantly limits the expressiveness of the encoding, scaling only linearly with the number of qubits. In contrast, encoding classical data via a unitary operator acting on a quantum state leverages the full expressive power of the Lie algebra $\mathfrak{su}(2^\eta)$, which offers up to $2^{2\eta} - 1$ free parameters. This yields an exponential increase in representational capacity (by a factor of $2^{\eta-1}$) over state-based encoding, offering significantly greater flexibility and expressivity for quantum learning tasks.

This insight motivates our approach: rather than embedding data directly into a quantum state, we build unitary operators from structured combinations of algebraic generators. Specifically, we construct a complete and well-behaved set of Hermitian generators that span a subalgebra of $\mathfrak{su}(2^\eta)$. These generators are derived systematically to ensure linear independence, closure under commutation, and coverage of all valid operator directions in the Hilbert space. The full derivation and construction procedure are detailed in Appendix A, including Alg. 2 outlining the algorithmic formulation. Overall, this construction yields a complete and hardware-compatible generator basis for building expressive and efficient quantum kernels, as formally stated in the following theorem.

**Theorem 3.1.** *Let $\mathfrak{H}$ be the set of Hermitian generators constructed as described in Alg. 2. Then $\mathfrak{H}$ spans a Lie subalgebra $\mathfrak{h} \subseteq \mathfrak{su}(2^\eta)$ that is closed under commutation, linearly independent, and expressible in terms of Pauli basis elements, ensuring both algebraic validity and implementability on quantum hardware.*

*Proof.* The generator set $\mathfrak{H}$ is constructed from three families of Hermitian matrices: off-diagonal real symmetric matrices (Eq. 13), off-diagonal purely imaginary anti-symmetric matrices (Eq. 14), and diagonal traceless real matrices (Eq. 15). Together, these matrices span the entire space of traceless Hermitian operators on $\mathbb{C}^{2^\eta}$, yielding $4^\eta - 1$ linearly independent elements, which matches the dimension of the Lie algebra $\mathfrak{su}(2^\eta)$. Their construction guarantees closure under the commutator operation, fulfilling the necessary condition for forming a Lie subalgebra $\mathfrak{h} \subseteq \mathfrak{su}(2^\eta)$. Moreover, since the Pauli basis forms a complete operator basis of $\mathfrak{su}(2^\eta)$, each Hermitian generator $h_k \in \mathfrak{H}$ can be decomposed as a real linear combination of Pauli basis elements. This establishes a direct correspondence between our generator-based formalism and standard quantum circuit implementations, where unitary operations are constructed from Pauli rotations. $\square$

This implies that any unitary generated by $h_k$ can, in principle, be synthesized into a gate-based quantum circuit using known decomposition techniques. Hence, $\mathfrak{H}$ not only provides complete algebraic coverage for encoding but also ensures practical compatibility with current quantum hardware. With the set of generators $\mathfrak{H}$ in hand, it is now possible to encode a real-valued $g$-dimensional parameter vector $\boldsymbol{\phi} \in \mathbb{R}^g$ into a sequence of unitary operators. Theoretically, for an 8-qubit system, we could encode $65,535$ parameters into this sequence. Therefore, we propose to

merge them into *Variational Generator Groups* (VGGs), combining multiple generators with a single parameter. To do so, we split the list of generators into equally sized partitions. Every group $\mathcal{G} \subset \mathfrak{H}$ constitutes a set of generators linearly combined to form a single Hermitian matrix $\hat{H}_i$, constructed according to Alg. 1, which can be mapped to a unitary operator $\hat{U}$ using the time development of the quantum state, substituting the time dependence by a parameter element $\phi_i$:

$$\hat{U}_{\phi_i} = e^{-i \cdot \phi_i \cdot \hat{H}_i} \tag{2}$$

---

**Algorithm 1** Construction of Variational Generator Groups (VGGs)

---

**Require:** Set of generators $\mathfrak{H}$, Number of groups $g \in [1, 2^{2\eta} - 1]$, Projection width $w \in [1, 2\eta]$
**Ensure:** Hermitian Matrix Groups $\hat{H}$
1: Initialize Hamiltonian matrix: $\hat{H} \leftarrow \mathbf{0}_{(g,H,H)}$
2: Generate generator to group mapping according to the specified width $w$:
3: $idx \leftarrow \langle (j \cdot 2^w \bmod |\mathfrak{H}|,\ j \bmod g)\ \mid\ j \in \{0, 1, \ldots, |\mathfrak{H}| - 1\}\rangle$
4: **for all** $i \in [0, g[\ , \forall (j, k) \in \text{enumerate}(idx)$ s.t. $j \bmod g = i$ **do**
5: $\quad \hat{H}[i][\mathfrak{H}[k][0]] \leftarrow \hat{H}[i][\mathfrak{H}[k][0]] + \mathfrak{H}[k][1]$     $\triangleright$ $[\cdot]$ is used for array indexing here
6: **end for**

---

By grouping the generators, we are able to define the number of parameters that are introduced into the unitary. We choose the total number of groups $g$ (i.e., the number of embeddable parameters) as:

$$g = \frac{|\mathfrak{H}|}{\Gamma_\eta}\ , \text{ with } \qquad \Gamma_\eta = \begin{cases} 2 \cdot \Gamma_{\eta-1} + 1 & \text{if } \eta > 2 \text{ and } \eta \text{ is odd,} \\ 2 \cdot \Gamma_{\eta-1} - 1 & \text{if } \eta > 2 \text{ and } \eta \text{ is even,} \\ 1 & \text{if } \eta \leq 2, \end{cases} \tag{3}$$

to ensure scaling the number of generators per group $\Gamma_\eta$ approximately exponentially with the total number of generators $|\mathfrak{H}|$. The grouping process is further parameterized by the projection width $w$ that determines the stride at which generators are assigned to groups. By varying this projection width, we can control the density of generators per VGG (cf. Fig. 4). Consequently, Alg. 1 ensures an even distribution of generators across all groups. Overall, using a wide stride (i.e., $w = 1$) helps promote linear independence among the grouped operators by mixing structurally diverse generators, which can improve parameter identifiability. Further justification for this approach, including comparisons to ungrouped and fully grouped schemes, is discussed in Appendix B, while the empirical properties of the resulting VGGs are analyzed in Sec. 6 and Appendix C. The theoretical analysis in Appendix D furthermore shows that expressivity is preserved as long as the groups form a strict partition of $\mathfrak{H}$.

## 4 QUANTUM GENERATOR KERNELS

Via this procedure, we create a VGG with a unitary operator $\hat{U}_{\phi_i}$ for every parameter element $\phi_i$, which can be decomposed into quantum gates or applied directly onto an initial quantum state. When multiplying the set of VGGs in a sequence, we obtain one condensed unitary incorporating all parameters $\phi$:

$$\hat{U}_\phi = \exp\left(\sum_{i=1}^{g} -i \cdot \phi_i \cdot \hat{H}_i\right) \tag{4}$$

As shown in Eq. (19), the group of unitary matrices is closed regarding multiplication; hence, $\hat{U}_\phi$ must also be unitary. We refer to this unified operator $\hat{U}_\phi$ as the *Quantum Generator Kernel* (QGK), illustrated in Fig. 1, which is applied via

$$\psi(x) = \hat{U}_x |\Psi\rangle = |\Psi'\rangle \tag{5}$$

to the totally mixed initial state $\langle\Psi|$, generated starting from the ground state $\langle 0|$ using $\otimes_{i=1}^{\eta} \mathcal{H} |0\rangle = |\Psi\rangle$, with $\eta$ qubits and the Hadamard gate $\mathcal{H}$ applied to all qubits. To calculate the kernel matrix $K$ from our QGK, we use the fidelity as the distance between the states:

$$K = k(x_i, x_j) = \left| \langle\Psi| \hat{U}_{x_j}^\dagger \hat{U}_{x_i} |\Psi\rangle \right|^2 \tag{6}$$

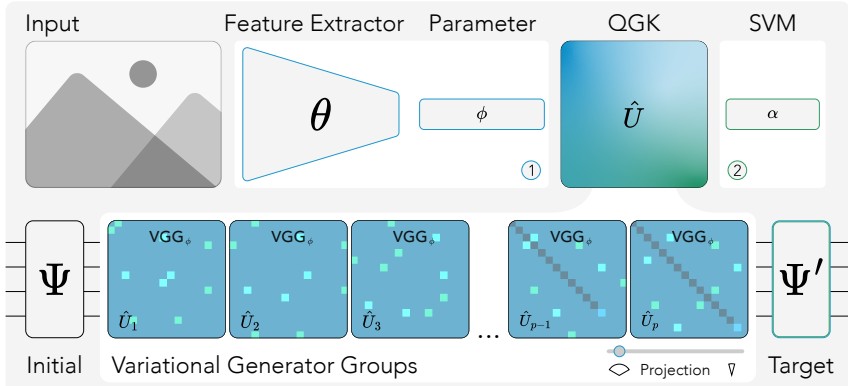

Figure 1: Quantum Generator Kernel: A generator-based quantum kernel architecture based on VGGs for parameterizable projection. Each colored matrix corresponds to one of $g = 51$ *Variational Generator Groups* (VGGs) merged for $\eta = 4$ qubits, visualized as heatmaps of the magnitude (blue) and phase (green) of the resulting generators merged into the operator. The QGK is parameterized by the context $\phi$, which is either given directly by the input or extracted from the input using a feature extractor, adapted during the pre-training phase (1) by updating the parameters $\theta$ to minimize the Kernel-Target Alignment (KTA) loss. In phase (2), a support vector machine (SVM), parameterized by $\alpha$, is trained using the resulting QGK $\hat{U}$.

To embed data using the QGK (cf. Fig. 1, phase 1), we can either use the input data to parameterize the VGGs directly, i.e., $\phi = x \in \mathbb{R}^d$ as denoted in Eq. (5), or use a feature extractor $\mathcal{F}_{\theta} : \mathbb{R}^d \mapsto \mathbb{R}^g$, parameterized by $\theta$, s.t. $\phi = \mathcal{F}_{\theta}(x)$. Note that to directly embed input data, the number of groups $g$ must be set according to the input dimension $d$. However, when using a feature extractor, we can decouple the input dimension from the number of generator groups. This enables dimensionality reduction, which we quantify via the compression factor $\gamma = d/g$. In our experiments, we restrict $\mathcal{F}_{\theta}$ to a linear affine transformation of the form $\phi = Wx + b$. This avoids introducing classical nonlinearities, and ensures that all expressive capacity stems from the quantum kernel. With the embedded data, the kernel matrix can be used to fit a support vector machine (SVM) parameterized by $\alpha$ according to Eq. (1), to perform arbitrary binary classification tasks. To pre-train the parameterization $\theta = \langle W, b \rangle$ of the feature extractor, we suggest using the Kernel Target Alignment (KTA) loss, as suggested in (Hubregtsen et al., 2022):

$$\mathcal{L}_{\text{KTA}} = 1 - \frac{\text{Tr}(\boldsymbol{K}\boldsymbol{Y})}{\|\boldsymbol{K}\|_{\text{F}} \cdot \|\boldsymbol{Y}\|_{\text{F}}}, \tag{7}$$

with the kernel matrix $\boldsymbol{K}$ and the classification targets $\boldsymbol{Y}$, where $\text{Tr}(\boldsymbol{A})$ is the trace of matrix $\boldsymbol{A}$ and $\|\boldsymbol{A}\|_{\text{F}}$ is the Frobenius norm. The resulting model class remains confined to the RKHS induced by the kernel $\mathcal{K}_{\phi}$ (see Appendix E for a theoretical analysis).

## 5 RELATED WORK

**Embedding Processes** To encode classical data into quantum systems, various strategies have been developed, ranging from fixed mappings such as *basis encoding*, where binary values $b \in \{0, 1\}$ are mapped to computational basis states $|b\rangle$, to more compact methods such as *amplitude encoding*, where a normalized vector $v \in \mathbb{R}^{2^n}$ is embedded directly into the amplitudes of a quantum state $|\Psi\rangle$. While amplitude encoding uses only $\eta$ qubits to represent exponentially large vectors, its practical use is limited due to costly state preparation and reduced kernel expressivity unless followed by complex unitaries (Sun et al., 2023; Schuld et al., 2021b; Schuld & Killoran, 2019; Huang et al., 2021). *Angle encodings*, or Pauli rotational embeddings, instead map real-valued inputs into single-axis rotations such as $R_X(x)$ or $R_Z(x)$, and form the basis of many variational quantum circuits. Their expressivity is often enhanced by *data reuploading* (Pérez-Salinas et al., 2020), which introduces nonlinearity through repeated injection of inputs, enabling the circuit to approximate Fourier-like transformations (Jaderberg et al., 2024). The Quantum Embedding Kernel (QEK) (Hubregtsen et al., 2022) leverages this mechanism to train kernel functions via KTA maximization. However,

such methods rely on axis-aligned encodings and fixed circuit structures. In contrast, recent work has explored more expressive multi-axis or multi-qubit rotational embeddings, often motivated by Fourier analysis or the algebraic structure of the Pauli group. In this context, our *Variational Generator Groups* (VGGs) can be seen as a structured and systematic *generalization* of multi-qubit angle encoding. Instead of encoding each input feature through a single-axis rotation, VGGs construct unitaries from grouped, algebraically structured combinations of Hermitian generators $\hat{H}_k = \sum_{h \in \mathcal{G}_k \subset \mathfrak{H}} h$, where each $h$ is a linear combination of Pauli strings spanning the full Pauli basis (cf. Theorem 3.1). The resulting unitary transformations $\exp(-i\phi_k \hat{H}_k)$ implement data-dependent, *multi-axis* and *multi-qubit* transformations within the Lie algebra $\mathfrak{su}(2^n)$. This distinguishes VGGs from traditional angle encoding in two key ways: (1) While angle encodings act on one Pauli axis per qubit, VGGs use grouped generators that cover structured subspaces of $\mathfrak{su}(2^n)$, enabling richer transformations per parameter. (2) Each group induces entangling, correlated rotations across qubits, going beyond independent, axis-aligned operations. Thus, VGGs form a principled extension of multi-axis angle encoding, bridging fixed rotational embeddings and full Hamiltonian variational models. This allows for greater expressivity per qubit, making it particularly effective for kernel learning in qubit-constrained settings.

**Hybrid QML** Motivated by the limited capabilities of current quantum hardware, hybrid quantum-classical machine learning approaches have been explored in literature, which add pre- and postprocessing layers to the quantum model (Mari et al., 2020). While this approach allows the exploration of problems that are beyond the capabilities of current quantum hardware by handling larger data sizes, it blurs the individual contributions of the classical and the quantum components to the overall solution quality (Altmann et al., 2023; Kölle et al., 2024). This issue is particularly pronounced in *Dressed Quantum Circuits* (DQC) (Mari et al., 2020), where nonlinear neural networks appear both before and after the quantum circuit. By the universal approximation theorem (Hornik et al., 1989), these classical networks can approximate arbitrary continuous functions even if the quantum circuit acts trivially, making it difficult to attribute improvements to quantum processing. In contrast, our QGK uses only an affine transformation to parameterize generator weights and no nonlinear post-processing, ensuring that expressivity is strictly governed by the quantum kernel (see Appendix E). A recent example combining the above principles is the *Hardware Efficient Embedding* (HEE) (Thanasilp et al., 2024), where input-dependent rotations are arranged in layered circuits interleaved with entangling gates (e.g., CNOT or CZ). These embeddings are expressive and compatible with near-term devices, but they can lead to exponentially concentrated kernel values and barren features unless carefully controlled. To adapt the input dimensionality to the limited number of qubits, the authors apply *principal component analysis* (PCA). Even though the input dimensionality of our proposed generator-based approach scales exponentially with the number of qubits, in contrast to the linear scaling of the HEE, its preprocessing denotes a similar approach to the linear layer we introduce. However, rather than static feature extraction, we further use this pre-processing to pre-train the projection of the kernel. Another recent direction is the *Projected Quantum Kernel* (PQK) (Huang et al., 2021), which avoids fidelity-based kernels by extracting quantum features through the *one-particle reduced density matrix* (1-RDM) and subsequently applying a classical kernel function. Similar to HEE, the PQK is limited by linear input scaling and relies heavily on classical compression techniques such as PCA. Furthermore, it may suffer when the underlying label structure is not well aligned with the kernel geometry induced by the projected quantum features. In contrast, the QGK supports high-dimensional inputs via grouped Hamiltonian encoding and offers kernel-target alignment (KTA) pre-training to adapt the kernel to task-specific structures.

**Generator-based approaches** The exploration of generator-based quantum computing is rather recent, compared to the longer exploration of gate-based variational circuits (Nielsen & Chuang, 2010). Generators are used in more mathematical explorations of quantum computing (Mansky et al., 2023a). In particular, they are found in the treatment of barren plateaus (Arrasmith et al., 2021; Goh et al., 2023; Ragone et al., 2024) and specialized circuits that focus on restricting subspaces (Schatzki et al., 2024; Nguyen et al., 2024). A direct equivalent to the generators does not exist in classical machine learning, owing to the different mathematical structure (Bronstein et al., 2021).

## 6 EMPIRICAL ANALYSIS

In this section, we aim to provide an empirical analysis of the resulting properties of the proposed *Quantum Generator Kernel* (QGK) and compare them to the *Quantum Embedding Kernel* (QEK), *Harware Efficient Embedding* (HEE), and *Projected Quantum Kernel* (PQK) as representative state-of-the-art approaches to quantum kernel methods, as well as classical *Radial Basis Function* (RBF) and Linear kernels. Fig. 2 summarizes our results.

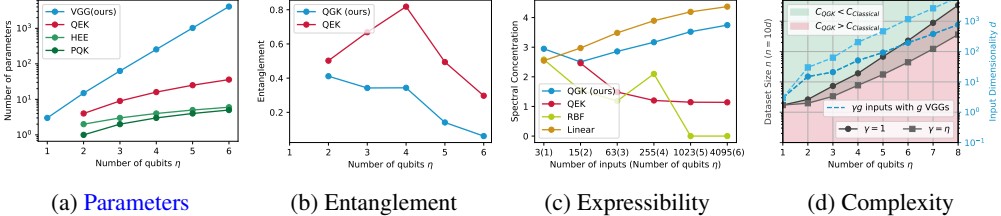

| (a) Parameters | (b) Entanglement | (c) Expressibility | (d) Complexity |

Figure 2: Comparing the scaling behavior of the QGK to classical and quantum kernels w.r.t. the number of available qubits $\eta$ (i.e., $4^\eta - 1$ input dimensions), regarding: (a) the number of available parameters, (b) the entanglement capability by means of the Meyer-Wallach measure, (c) the expressibility by means of the spectral concentration, and (d) the computational complexity, showing the complexity breakeven dataset size $n$ when classically simulating the QGK (left y-axis) and the number of VGGs and resulting inputs (right y-axis, blue), scaled to $n = 10d$ to satisfy $n \gg d$.

**Parameter efficiency** Fig. 2a shows the scalability of the number of parameters w.r.t. the number of qubits. For the QEK, we assume a maximum of $\eta^2$ input parameters, i.e., using $\eta$ layers to embed the input data, scaling quadratically with the number of qubits. Utilizing generators in Hilbert space rather than fixed gate-based embedding processes, our approach shows significantly improved qubit efficiency, scaling exponentially with the number of qubits. Thus, QGK offers an overall higher data capacity, which is especially crucial for realizing real-world applications on near-term quantum devices. To ensure a fair comparison with equal parameter counts, the following analysis is based on the maximum number of QGK parameters, where the QEK is extended by parameterized reuploading layers to replenish the additional parameters. As mentioned above, the input capacity of methods such as HEE and PQK scales only linearly with the number of qubits. Therefore[1], we omit them from the following analysis of the theoretical kernel properties, though we include them in subsequent empirical comparisons.

**Kernel properties** Fig. 2b shows the maximum entanglement capabilities by means of the Meyer-Wallach measure. Here, the embedding-based approach shows higher capabilities, due to the entanglement layers. For both approaches, the entanglement capability peaks at lower qubit numbers and steadily decreases afterwards. However, given that strong entanglements might also impede optimization, this property could actually be beneficial to both approaches. Fig. 2c compares the spectral concentration of various kernels as a measure of expressibility. The QGK shows a clear upward trend with increasing qubit count, benefiting from its generator-based structure and high parameter density. In contrast, the QEK exhibits decreasing expressibility, suggesting weaker scalability with added parameters. The classical RBF kernel also shows decreasing spectral concentration with increasing input size, while the Linear kernel, similar to QGK, increases steadily. Notably, QGK maintains lower spectral divergence than the Linear kernel across input sizes despite operating in exponentially larger Hilbert spaces, indicating strong resilience against expressibility collapse and exponential concentration. These results highlight QGK's favorable expressibility-to-learnability trade-off, making it a scalable and efficient alternative for hybrid quantum-classical pipelines. To assess architectural sensitivity, we analyze the impact of group size and projection stride in Appendix C. The results confirm that expressibility and entanglement remain stable across all tested configurations, with exponential group scaling and wide stride (w = 1) yielding the most consistent and expressive feature

---

[1]Matching the input-scale of our QGK would require tens to thousands of qubits, which is infeasible for simulation or current hardware without aggressive classical dimensionality reduction. Such hybrid compression, however, obscures the intrinsic quantum embedding capabilities being evaluated.

maps. Finer groupings enhance expressiveness slightly, while projection width has only marginal impact. These findings reinforce the theoretical robustness guarantees presented in Appendix D and support our default grouping design as a scalable and well-conditioned choice.

**Computational Complextiy**    Despite the exponential scaling in qubit number $\eta$, the QGK remains classically simulable due to its decomposition into tensor-efficient operations: generator construction, input projection, quantum evolution, and pairwise kernel evaluation. Unlike classical kernels such as RBF or Linear, which incur $\mathcal{O}(n^2 \cdot d)$ cost for computing the similarity matrix, QGK scales as $\mathcal{O}(4^\eta + n \cdot \gamma \cdot g^2 + n \cdot 8^\eta + n^2 \cdot 2^\eta)$, with $g$ VGGs and the compression ratio $\gamma = d/g$. This structure favors sample-efficient scenarios revealed by the complexity analysis shown in Fig. 2d, indicating two key properties of the QGK: (i) for low qubit number ($\eta \leq 5$) QGK offers a computational advantage over classical kernels when using a 1:1 mapping ($\gamma = 1$). (ii) to enable larger-scale applications (e.g., $d > 100$) efficiently, hybrid approaches with $\gamma > 1$ are required to compress the input. E.g., using $\gamma = \eta$ pushes the lower efficiency bound further down, such that the input dimension reference ($d = \eta g$ for $g$ VGGs, light blue) does not intersect anymore. This hybrid execution enables scalable and classically feasible QGK training in the NISQ era and lays the groundwork for full quantum execution on future fault-tolerant architectures. A detailed analysis, including complexity thresholds, approximations, and formal bounds, is provided in Appendix F. Importantly, while the QGK provides a path to handling large input dimensionality efficiently it shares the same quadratic complexity in the number of training samples $n$ as all classical kernel methods due to the $\mathcal{O}(n^2)$ kernel matrix computation. As with classical approaches, this can become prohibitive for very large datasets. To mitigate this, kernel approximation techniques such as the Nyström method (Williams & Seeger, 2000) or random feature expansions (Rahimi & Recht, 2007) can be used to reduce training complexity to near-linear in $n$, with minimal performance degradation.

## 7    EVALUATION

**Setup**    To empirically validate the properties and advantages discussed above, this section demonstrates the kernels' trainability, scalability, and hardware applicability across various tasks. We use the `moons` and `circles` datasets from (Pedregosa et al., 2011), with $d = 2$ input features each, both augmented with 20% noise and the `bank` (Moro et al., 2014a;b) dataset with $d = 16$ input features as small-scale synthetic and real-world benchmarks with $n = 200$, and the 10-class `MNIST` (LeCun et al., 1998) ($d = 784$) and `CIFAR10` (Krizhevsky et al., 2009) ($d = 3072$) datasets with $n = 1000$, to demonstrate scalable real-world applicability. In addition to the QEK (Hubregtsen et al., 2022), PQK (Huang et al., 2021), and HEE (Thanasilp et al., 2024), we evaluate *Radial Basis Functions* (RBF), *Linear Kernels,* and a small *Multi-Layer Perceptron* (MLP) as classical state-of-the-art baselines. To ensure approximately even capabilities, the QEK and HEE circuits comprise one data-reuploading layer, i.e., $2 \cdot d/\eta$ layers parameterized by Y- and CZ-rotations for the QEK, and 2 X-rotational input embedding layers for the HEE. For the PQK baseline, we follow Huang et al. (2021), projecting to $\eta - 1$ input features via PCA and computing an RBF kernel over the one-particle reduced density matrix (1-RDM). Relabeling is omitted to reflect general-purpose scenarios. To parameterize the feature projection of the QGK, we use a single linear layer. The baseline MLP is using a single hidden layer of size $g$, matching the number of generator groups. To provide an ablation quantifying the impact of using KTA to train the kernel, we also report the untrained QGK performance (*QGK Static*). To additionally quantify the impact of the VGG-based kernel itself and delimit it from the classical preprocessing, we adapted our linear pre-processing to the classical Linear Kernel (*Linear KTA*) and HEE (*HEE Linear*), whereas *HEE* refers to the untrained approach using PCA for feature extraction. We pre-train the embedding parameterization and baseline MLP for 100 epochs using the Adam optimizer with learning rate $10^{\eta-1}$. Unless stated otherwise, we use $\eta = 2$ for the binary and $\eta = 5$ for the multi-class benchmarks. As a general performance metric, we use the classification accuracy on an unseen $10\%$ test-split, additionally reporting the *Kernel Target Alignment* (KTA). All results are averaged over eight random seeds with $95\%$ confidence intervals. Non-pretrained approaches are shown as dashed horizontal lines. All evaluations were conducted on Apple M2 Ultra hardware with 192GB memory and a total compute time of approximately 96 hours using torchquantum (Wang et al., 2022) [2].

---

[2]The required implementations are appended and will be open-sourced upon publication.

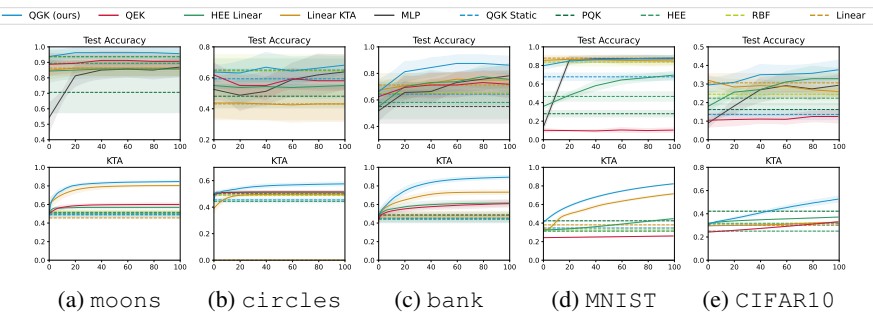

(a) `moons`   (b) `circles`   (c) `bank`   (d) `MNIST`   (e) `CIFAR10`

Figure 3: Training performance of the QGK(blue), QEK(red), PQK(dark green), HEE(green), RBF(yellow), and Linear(orange) Kernels and MLP(grey) w.r.t. the Test Accuracy (top) and KTA (bottom) in the `moons`, `circles`, `bank`, `MNIST`, and `CIFAR10` benchmarks, with the QGK outperforming all compared approaches.

**Binary Classification**   The training results for the `moons` dataset are shown in Fig. 3a. Notably, QGK outperforms both embedding-based approaches. Training the projection increases the final test accuracy from $94\%$ to $96\%$, and even outperforms both classical approaches, with a final KTA above $0.8$. Similar performance trends are prevalent in the `circles` benchmark shown in Fig. 3b, where the QGK outperforms all compared approaches, including both quantum and classical baselines. The overall higher accuracy compared to the HEE Linear ablation further underscores the effectiveness of our generator-based kernel over embedding-based alternatives. Looking at the results for the real-world `bank` dataset in Fig. 3c, the QGK continues to expand its performance lead. Despite the increased complexity introduced by the 16-dimensional input space, it achieves a mean final test accuracy of $86\%$, significantly outperforming all competing methods. This highlights QGK's ability to maintain robustness and accuracy even in higher-dimensional, non-trivial learning tasks.

**Large-Scale Tasks**   To evaluate the scalability of the Quantum Generator Kernel (QGK), we investigate its performance on the 10-class `MNIST` and `CIFAR10` benchmarks, shown in Fig. 3d and Fig. 3e. On `MNIST`, QGK achieves a final test accuracy of around $88\%$, matching the performance of the best classical method (Linear), and significantly outperforming QEK and HEE. Even without any trained projection, QGK Static achieves strong performance, highlighting the expressive nature of the generator-based representation. On `CIFAR10`, a considerably more challenging dataset ($d = 3072$), QGK still achieves the highest accuracy among all tested kernels around $0.38\%$, despite the strong compression required for five-qubit compatibility. In contrast, both quantum baselines (QEK, HEE) and classical kernels (RBF, Linear) show limited performance under the same constraints. These results illustrate the strong adaptability of QGK under large input dimensionality and its ability to generalize beyond synthetic or low-dimensional tasks.

**Hardware Compatibility**   To assess compatibility with current quantum hardware and evaluate robustness to noise, we compiled all compared circuits to IBM's Falcon architecture and simulated them using realistic noise models[3]. As summarized in Tab. 1, for the binary classification tasks, all methods maintain manageable depths below 100, allowing realistic simulation under noise. Notably, QGK outperforms all compared approaches (including both classical ones) even when executed under realistic hardware noise. On larger-scale tasks, the compiled depths diverge, reflecting significant differences in the encoding strategies (cf. Tab. 6). HEE achieves the lowest depth but only encodes five features (less than 1% of the input dimension, even for the lower-dimensional `MNIST`), relying heavily on classical preprocessing, arguably limiting its comparability. At the other extreme, QEK encodes all features without classical reduction, but at the cost of unmanageable circuit depths exceeding 20k, demonstrating its poor scalability due to the fixed embedding structure. In contrast, QGK strikes a practical balance: grouping generators to embed 93 features using five qubits yields manageable depths below 5k, without heavy preprocessing. While such depths remain intractable for today's noisy hardware, exhaustive simulation offers limited value and is better reserved for future error-corrected devices. Notably, QGK's structured design allows further depth reductions via generator pruning, paving the way for efficient deployment on fault-tolerant quantum systems.

---

[3]We use the simulated 5-qubit IBM Falcon processor *FakeQuito* (Javadi-Abhari et al., 2024).

| Dataset ($d$, $\eta$) | QGK (ours) | QEK | HEE | RBF | Linear |
|---|---|---|---|---|---|
| moons (2, 2) | $\mathbf{\textit{0.96} \pm \textit{0.04}(\textit{28})}$ | $\textit{0.91} \pm \textit{0.05}(\textit{50})$ | $\textit{0.89} \pm \textit{0.06}(\textit{18})$ | $0.93 \pm 0.04$ | $0.86 \pm 0.05$ |
| circles (2, 2) | $\mathbf{\textit{0.69} \pm \textit{0.07}(\textit{28})}$ | $\textit{0.58} \pm \textit{0.09}(\textit{50})$ | $\textit{0.64} \pm \textit{0.06}(\textit{18})$ | $0.64 \pm 0.11$ | $0.43 \pm 0.10$ |
| bank (16, 2) | $\mathbf{\textit{0.87} \pm \textit{0.06}(\textit{28})}$ | $\textit{0.72} \pm \textit{0.10}(\textit{380})$ | $\textit{0.61} \pm \textit{0.10}(\textit{18})$ | $0.66 \pm 0.09$ | $0.71 \pm 0.09$ |
| MNIST (784, 5) | $\mathbf{0.88 \pm 0.03(4754)}$ | $0.10 \pm 0.02(24084)$ | $0.41 \pm 0.03(53)$ | $0.84 \pm 0.04$ | $\mathbf{0.88 \pm 0.03}$ |
| CIFAR10 (3072, 5) | $\mathbf{0.38 \pm 0.05(4754)}$ | $0.09 \pm 0.01(94485)$ | $0.21 \pm 0.03(53)$ | $0.24 \pm 0.03$ | $0.31 \pm 0.05$ |

Table 1: Final test accuracies across five benchmarks, comparing QGK (ours) with quantum (QEK, QGK Static, HEE) and classical (Linear, RBF) kernels. Italic values indicate results obtained from noisy circuit simulation on 5-qubit IBM Falcon hardware (Quito), with the compiled circuit depth in parentheses. Even under hardware noise, QGK consistently achieves the highest accuracy, demonstrating superior robustness and scalability across both synthetic and real-world datasets.

## 8 CONCLUSION

In this paper, we introduced *Quantum Generator Kernels* (QGK), a novel generator-based approach to quantum kernel methods. The QGK consists of Variational Generator Groups (VGGs) that merge a set of universal generators into parameterizable groups. Building upon universal generators in Hilbert space, QGKs offer significantly improved parameter scalability compared to common gate-based approaches, employing fixed embedding processes. Empirical studies across five benchmarks demonstrate that the QGK achieves superior trainability and classification accuracy, consistently outperforming quantum baselines and matching or exceeding the best classical methods, even under realistic hardware noise.

**Key Results** On both synthetic binary tasks, QGK outperforms classical and quantum embedding-based methods, showcasing strong expressiveness under limited quantum resources. On the real-world bank dataset, QGK maintains a clear lead, reaching $87\%$ accuracy despite the 16-dimensional input space. On the larger-scale MNIST benchmark, QGK reaches 88% accuracy, matching the best classical kernel (Linear) and significantly outperforming all quantum baselines. On the more complex CIFAR10, QGK achieves 38%, clearly surpassing QEK (9%), HEE (21%), and even classical kernels like RBF (24%) and Linear (31%). These results highlight QGK's strong scalability, embedding efficiency, and superior learning capacity under high-dimensional input conditions. Despite limited qubit budgets, QGK provides expressive embeddings through generator-grouped unitaries, offering a practical trade-off between depth, accuracy, and embedding richness.

**Limitations** Although generator-based quantum kernels provide strong theoretical expressiveness and favorable scaling, they are not natively supported on current hardware. Combined with today's limited quantum device capabilities, this constrains their immediate large-scale deployment. At the same time, this early stage presents an opportunity for application-driven co-design that may enable native execution of generator-based models in the future. To assess near-term viability, we compiled QGK circuits to current IBM hardware and simulated them under realistic noise. On small-scale tasks, compiled depths remain well below 100 gates, confirming competitive feasibility relative to conventional gate-based methods. Notably, QGK maintains leading accuracy over both classical and quantum baselines even under noise for all evaluated tasks. For large-scale datasets, however, compiled depths exceed the capabilities of current noisy hardware. Here, efficient tensor-based implementations combined with compression provide competitive classical execution until fault-tolerant quantum devices capable of handling large-scale embeddings become available. Additional efficiency gains may be achieved through generator pruning, further reducing depth.

**Outlook** For small- and medium-scale tasks, hybrid execution offers a practical path: classical preprocessing can reduce dimensionality before quantum embedding, enabling robust performance on today's noisy devices. For large-scale datasets such as MNIST or CIFAR-10, efficient tensor-based implementations provide a tractable classical alternative, keeping generator-based kernels competitive until quantum hardware matures and even outperforming classical baselines like RBF and Linear. In the long term, with the advent of fault-tolerant systems, QGK could be executed fully quantum, including variational training of projections and native generator-based operations. Overall, QGK provides a scalable, expressive, and classically efficient kernel method for near-term hybrid deployment, while paving the way toward a generator-based paradigm of quantum-native learning in future hardware generations.

ETHICS STATEMENT

This work adheres to the ICLR Code of Ethics. It does not involve human subjects, personal data, or sensitive information. Large language models (LLMs) were used solely to assist with language editing and structuring; all technical content and results were developed and verified independently by the authors. We anticipate that advances in quantum machine learning, and in particular the new paradigm of generator-based quantum kernel methods introduced here, may broaden the applicability of kernel learning and enable scalable use of quantum resources in future AI systems.

REPRODUCIBILITY STATEMENT

We have taken several measures to ensure reproducibility of our results. A detailed description of the proposed Quantum Generator Kernel (QGK) method, including generator construction, grouping procedure, and training pipeline, is provided in the main text and Appendix A, B, D, and E. Theorems and proofs of the algebraic properties are included in Appendix F. Hyperparameters, datasets, and experimental setups are reported in Sec. 7 and Appendix C and summarized in Tables 6 and 1. Noise simulations and hardware compilation details are given in Appendix G, with compiled depths explicitly reported. All datasets used (moons, circles, bank, MNIST, and CIFAR10) are publicly available, with preprocessing steps documented in the supplementary materials. For robustness, we report averages over eight random seeds. The full implementation is uploaded in the code appendix for reproducibility during review and will be open-sourced software upon publication.

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

## A  Deriving a Universal set of generators

The SU(N) group includes all unitary $N \times N$ matrices under multiplication. As a Lie group, the group elements form a smooth manifold and tangential space, the Lie algebra $\mathfrak{su}(N)$. The Lie algebra itself is defined as an additive real vector space characterized together with a commutator relation, satisfying the Jacobi identity. Further, it exhibits the same dimension as the respective Lie group. The direct connection between elements of a Lie group G and elements of the corresponding Lie algebra $\mathfrak{g}$ can be defined by:

$$\forall_{X \in \mathfrak{g}} \quad e^{-t \cdot X} \in G \quad \text{with } t \in \mathbb{R} \tag{8}$$

However, $X \in \mathfrak{g}$ just holds as long as $\forall_{t \in \mathbb{R}} e^{-t \cdot X} \in G$. The Lie algebra itself constitutes a vector space which is spanned by a number of base elements $e_i$, while $i$ denotes the respective dimension. The number of base elements is equivalent to the dimension of the Lie algebra. Using the fact that the Lie group G represents a differentiable manifold, the associated Lie algebra $\mathfrak{g}$ represents the tangent space to G at its identity element. This picture leads to the relation given in Eq. (9), while the indices $l$ and $k$ denote the application of the equation on a matrix element at the respective position.

$$\left. \frac{\partial}{\partial x_i} A_{lk}(x_1, x_2 ... x_n) \right|_{x_1=0, x_2=0 ... x_n=0} = (e_i)_{lk} \quad A \in G \tag{9}$$

Since it can be shown that the set of traceless square anti-Hermitian $N \times N$ matrices constitutes the $\mathfrak{su}(N)$ algebra, a complete base set $\{e_1, e_2 ...\}$ of linear independent representatives of $\mathfrak{su}(N)$ is sufficient to create any element within this algebra. By making use of the relation given in Eq. (8), we can map any element of $\mathfrak{su}(N)$ to an element of SU(N). Additionally, these anti-Hermitian base elements $e_i$ can be converted into so called Hermitian generator elements $h_i$ by using:

$$e_i = -\frac{1}{2} i \cdot h_i \tag{10}$$

Although directly related to the base elements, the generators do not form a basis of the Lie algebra by themselves. However, in order to create a base element out of a generator, we simply convert Eq. (10).

The connection between algebra and group via the algebra is exact (Mansky et al., 2024). It can also be used to build quantum circuits from Lie algebra elements, as every element of the Lie group has a corresponding quantum circuit element. In general, this representation of the group element as a quantum circuit is difficult to find and generally requires $4^\eta$ operations to represent (Bergholm et al., 2005; Shende et al., 2005; Mansky et al., 2023a). With the choice of a particular basis, this approach can be simplified. The Pauli basis $\Pi = \bigotimes_i^\eta \{\sigma_x, \sigma_y, \sigma_z, \mathbb{I}\} \backslash \mathbb{I}^\eta$ is the discrete group of Pauli strings, the tensor product of Pauli matrices. This restricts the Lie algebra dimension to $N = 2^\eta$, the natural scaling of qubit-based quantum computers.

The Lie algebra elements can be expanded to quantum circuits mechanistically (Mansky et al., 2023b).

Due to the fact that the group of traceless square anti-Hermitian $N \times N$ matrices constitutes the $\mathfrak{su}(N)$ algebra, a complete set of base elements can be easily formulated. Through the multiplication with a complex factor, Hermitian generators can be constructed following Eq. (10). This also makes it possible to start directly by creating a set of base elements for Hermitian matrices (generators) and convert them back to anti-Hermitian matrices by applying the factor. This can be done during the transformation into a unitary displayed in Eq. (4).

Firstly, the set of generators $\mathfrak{H}$ needs to show linear independence among all its elements. This can be satisfied if every generator contains at least one element $a_{lk} \neq 0$ for which all other generators show $a_{lk} = 0$. To further ensure that the found generators actually form a basis for Hermitian matrices, any pair of generators $h_i$ and $h_j$ needs to satisfy the following condition:

$$\text{Tr}(h_i, h_j) = 2 \cdot \delta_{ij} \tag{11}$$

Finally, it needs to be ensured that all generators $h_i$ and $h_j$ fulfill the commutator condition characterizing all basis elements of a Lie algebra:

$$[h_i, h_j] = \sum_{k=1}^{n} C_{ikl} \cdot 2i \cdot h_k \quad with \ \forall_{k \in \{1, 2..n\}} h_k \in \mathfrak{H} \ \wedge \ \forall_{i,k,l} C_{ikl} \in \mathbb{R} \tag{12}$$

Based on these three conditions, the following three generator subsets given in Eqs. (13), (14), and (15) can be found. Here, the first subset defines the off-diagonal real elements, the second one the off-diagonal imaginary elements, and the third set the real diagonal elements.

$$
\begin{pmatrix} 0 & 1 & & & 0 \\ 1 & 0 & & & \\ & & \ddots & & \\ 0 & & & & 0 \end{pmatrix}, \ldots,
\begin{pmatrix} 0 & 0 & & & 1 \\ 0 & 0 & & & \\ & & \ddots & & \\ 1 & & & & 0 \end{pmatrix}, \ldots,
\begin{pmatrix} 0 & & & & 1 \\ & \ddots & & & \\ & & & 0 & 1 \\ 0 & & & 1 & 0 \end{pmatrix}
\tag{13}
$$

$$
\begin{pmatrix} 0 & -i & & & 0 \\ i & 0 & & & \\ & & \ddots & & \\ 0 & & & & 0 \end{pmatrix}, \ldots,
\begin{pmatrix} 0 & 0 & & & -i \\ 0 & 0 & & & \\ & & \ddots & & \\ i & & & & 0 \end{pmatrix}, \ldots,
\begin{pmatrix} 0 & & & & 0 \\ & \ddots & & & \\ & & & 0 & -i \\ 0 & & & i & 0 \end{pmatrix}
\tag{14}
$$

For the construction of the generators containing diagonal elements, the traceless condition must be regarded. This leads to the construction of a set of matrices given in Eq. (15).

$$
\begin{pmatrix} 1 & 0 & & & 0 \\ 0 & -1 & & & \\ & & \ddots & & \\ 0 & & & & 0 \end{pmatrix}, \ldots,
\frac{1}{\sqrt{(n-1)!}}
\begin{pmatrix} 1 & 0 & & & 0 \\ 0 & 1 & & & \\ & & \ddots & & \\ 0 & & & & -(n-1) \end{pmatrix}
\tag{15}
$$

Algorithmically, the set of generators can be constructed with computational complexity $\mathcal{O}(4^\eta)$ via:

---

**Algorithm 2** Construction of Generators

---

**Require:** Number of qubits $\eta$
**Ensure:** Set of generators $\mathfrak{H}$
1: Compute Hilbert space dimension: $H \leftarrow 2^\eta$
2: Initialize empty set $\mathfrak{H}$ and append generators in the format $((\boldsymbol{r}, \boldsymbol{c}), \boldsymbol{v})$,
    where $|\boldsymbol{r}| = |\boldsymbol{c}| = |\boldsymbol{v}|$, $\boldsymbol{r}$ and $\boldsymbol{c}$ are lists of indices in the Hamiltonian matrix,
    and $\boldsymbol{v}$ is the list of corresponding (non-zero) values of the respective generator.
3: $\mathfrak{H} \leftarrow \mathfrak{H} \cup \{((\langle r,c\rangle, \langle c,r\rangle), \langle 1+0j, 1+0j\rangle) \mid r \in [0, H[ , \ c \in ]r, H[ \}$
4: $\mathfrak{H} \leftarrow \mathfrak{H} \cup \{(((r,c),(c,r)),(0+1j,0-1j)) \mid r \in [0, H[ , \ c \in ]r, H[ \}$
5: $\mathfrak{H} \leftarrow \mathfrak{H} \cup \{(((0,\ldots,j),(0,\ldots,j)),((1/\sqrt{i!})_i \cup \{-j/\sqrt{i!}\})) \mid j \in [1, H[ \}$

---

Fig. 4 shows a visual comparison between wide and narrow projections widths.

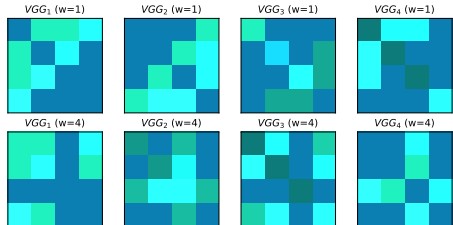

Figure 4: Projection heatmaps for VGGs for $\eta = 2$ qubits, merged in $g = 4$ groups with widths $w = 1$ (upper) and $w = 4$ (lower), visualizing the magnitude (blue) and phase (green) of the resulting generator matrices $\hat{\boldsymbol{H}}$.

## B ALTERNATIVE GROUPING APPROACHES

One alternate approach to form a single unitary matrix incorporating all $2^{2 \cdot \eta} - 1$ parameters assigns every free parameter to a specific generator of the underlying $\mathfrak{su}(N)$ algebra. By forming one large linear combination of generators $h_i$ multiplied by the respective parameters $\phi_i$, one Hermitian operator $\hat{H}$ is produced:

$$\hat{H} = \sum_{i=1}^{2^{2 \cdot \eta} - 1} \phi_i \cdot h_i \quad , \tag{16}$$

while $\eta$ again corresponds to the number of qubits used. This Hermitian operator $\hat{H}$ can then be converted to a single unitary matrix $\hat{U}$:

$$\hat{U} = e^{-i\hat{H}} = e^{-i \sum_{i=1}^{2^{2 \cdot \eta} - 1} \phi_i h_i} \tag{17}$$

An important disadvantage of this approach, however, is the inability to form a finite expression of the derivative of $\hat{U}$ with regard to one of the involved free parameters $\phi i$ in the general case. This issue becomes apparent when differentiating $\hat{U}$ while displaying the involved exponential function in its series representation. The difficulties in forming a finite expression for the derivative originate from the non-commuting characteristic present between specific generators $h_i$ and $h_j$. An example of this is given in Eq. (18):

$$\hat{H} = \sum_{i=1}^{2^{2 \cdot \eta} - 1} \phi_i \cdot h_k \quad \forall \quad h_k \in \mathcal{G}_i \tag{18}$$

To circumvent the problem, we chose another approach, which enables the formulation of a finite expression of the derivative and which is described in Sec. 3 in more detail. If we separate the set of generators into clusters, assigning one free parameter to every cluster respectively, we generate several unitary sub-operators $\hat{U}_i$, which we contract to a single one according to Eq. (4). Since every sub-operator contains only one free parameter, no commutation is necessary in order to form a finite derivative; hence, the non-commuting characteristic does not cause any problems. Also, as given by the following proof showing that the unitary matrices $U$ representing the transformation of a quantum state form a closed group regarding matrix multiplication, the resulting operator is also unitary:

$$\begin{aligned} \hat{U} = \hat{U}_{\phi_1} \hat{U}_{\phi_2} \dots \hat{U}_{\phi_p} \quad & \cap \quad \forall_{i \epsilon \{1 \dots p\}} \ \hat{U}_{\phi_i}^+ \hat{U}_{\phi_i} = \mathbb{1} \\ \Longrightarrow \hat{U}^+ \hat{U} &= (\hat{U}_{\phi_1} \hat{U}_{\phi_2} \dots \hat{U}_{\phi_p})^+ (\hat{U}_{\phi_1} \hat{U}_{\phi_2} \dots \hat{U}_{\phi_g}) \\ &= \hat{U}_{\phi_g}^+ \dots \hat{U}_{\phi_2}^+ \hat{U}_{\phi_1}^+ \cdot \hat{U}_{\phi_1} \hat{U}_{\phi_2} \dots \hat{U}_{\phi_g} \\ &= \hat{U}_{\phi_g}^+ \dots \hat{U}_{\phi_2}^+ \cdot \mathbb{1} \cdot \hat{U}_{\phi_2} \dots \hat{U}_{\phi_g} \\ &= \mathbb{1} \end{aligned} \tag{19}$$

However, maximizing the number of parameters incorporated in $\hat{U}$, i.e., introducing one parameter per generator, requires clusters containing only a single generator, respectively. As a result, we would create one sub-operator per generator. This means we have to do a matrix multiplication for every additional generator in order to get the contracted unitary $\hat{U}$. This again sets a challenge to the approach since the generator number increases exponentially ($2^{2 \cdot \eta} - 1$) with $\eta$ qubits, making the calculation of $\hat{U}$ computationally expensive. This justifies the implemented flexibility of choosing the size of the used clusters, enabling the adjustment of the tradeoff between computational expense and size of the introduced context.

To additionally reduce the computational complexity of the resulting operators, we furthermore utilize the following approximation:

$$\hat{U}_\phi = \exp\left( \sum_{i=0}^{g} -i \cdot \phi_i \cdot \hat{H}_i \right) \approx \prod_{i=0}^{g} \exp\left( -i \cdot \phi_i \cdot \hat{H}_i \right) = \prod_{i=0}^{g} \hat{U}_{\phi_i} \tag{20}$$

## C GROUPING HYPERPARAMETER ANALYSIS

The *Variational Generator Group* (VGG) construction (Alg. 1) introduces two key hyperparameters: (1) the number of generators per group, controlled by the number of groups $g$ (cf., Eq. 3), and (2) the projection width $w$, which determines how generators are assigned to groups (wide, medium, or narrow stride). This section analyzes how these hyperparameters influence the theoretical properties of the QGK, including entanglement capacity and expressibility (Fig. 5), compiled depth (Tab. 3), and their empirical impact on downstream learning performance (Tab. 4).

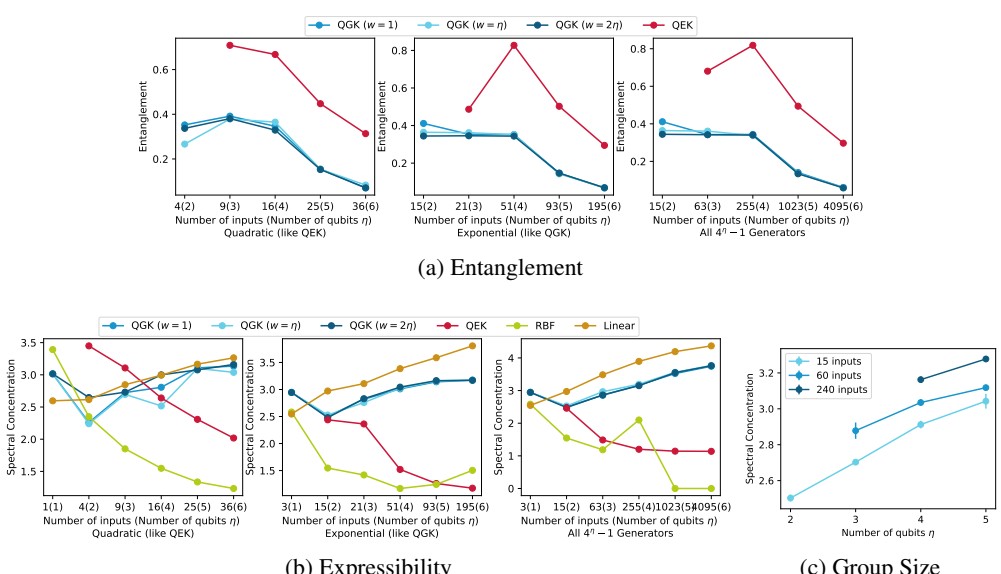

(a) Entanglement

(b) Expressibility       (c) Group Size

Figure 5: VGG grouping analysis and comparison regarding (a) the entanglement capability by means of the Meyer-Wallach measure and (b) the expressibility by means of the spectral concentration, accross three input scalings: *quadratic* ($g = \eta^2$, left) yielding the most generators per group, *exponential* (middle) with the number of generators per group chosen according to Eq. 3, and *all* (right) using the maximum available input dimensions, i.e., the total number of generators $4^\eta - 1$ and three projection widths: the default $w = 1$ causing a wide stride, i.e., assignment of distant generators to groups, $w = \eta$, causing a medium stride, and $w = 2\eta$, causing a narrow stride. (c) shows a comparison of spectral concentration across group sizes $g \in [15, 60, 240]$, averaged over projection widths ($w \in \{1, \eta, 2\eta\}$), with error bars indicating the standard deviation.

**Entanglement Capacity** Fig. 5a shows that QGK maintains stable entanglement capabilities across all examined projection widths and grouping strategies. Variance decreases with increasing numbers of qubits and groups, demonstrating robustness as system size grows.

**Expressivity** Fig. 5b shows the epxressibility of the QGK behaves overall robustly across different projection widths and numbers of qubits, particularly when using the suggested exponential scaling. e Finally, Fig. 5c summarizes the effect of different grouping configurations on kernel expressivity, showing three key insights: (1) QGK expressivity is highly robust across projection widths, with only minor variation between $w \in \{1, \eta, 2\eta\}$, even as the number of qubits and group sizes change. (2) Expressivity increases systematically with system size, reflecting the growing representational capacity of larger generator sets. (3) Finer groupings (i.e., more groups with fewer generators per group) consistently yield higher expressivity, indicating that distributing generators more granularly across VGGs enhances the diversity of attainable unitary transformations.

**Compiled Circuit Depth** Tab. 3 reports the compiled depths on IBM Falcon (*FakeQuito*, $\eta \leq 5$). Across all projection widths and input-scaling rules, depth variations are negligible. This is expected because the underlying set of generators remains identical; only their groupings differ. Thus, VGG rearrangements do not materially affect the gate count for fixed hardware constraints.

| Scaling | Width | $\eta = 1$ | $\eta = 2$ | $\eta = 3$ | $\eta = 4$ | $\eta = 5$ | $\eta = 6$ | Total ↓ |
|---|---|---|---|---|---|---|---|---|
| | | 1 inputs | 4 inputs | 9 inputs | 16 inputs | 25 inputs | 36 inputs | - |
| Quadratic (like QEK) | 1 | 0.00 | 0.12 | 0.01 | 0.03 | 0.01 | 0.02 | 0.20 |
| | $\eta$ | 0.00 | 0.14 | 0.02 | 0.26 | 0.01 | 0.07 | 0.50 |
| | $2\eta$ | 0.00 | 0.27 | 0.01 | 0.23 | 0.02 | 0.05 | 0.57 |
| | | 3 inputs | 15 inputs | 21 inputs | 51 inputs | 93 inputs | 195 inputs | - |
| Exponential (like QGK) | 1 | 0.00 | 0.00 | 0.02 | 0.02 | 0.01 | 0.00 | 0.05 |
| | $\eta$ | 0.00 | 0.02 | 0.04 | 0.00 | 0.00 | 0.00 | 0.08 |
| | $2\eta$ | 0.00 | 0.02 | 0.03 | 0.02 | 0.01 | 0.00 | 0.08 |
| | | 3 inputs | 15 inputs | 63 inputs | 255 inputs | 1023 inputs | 4095 inputs | - |
| All $4^\eta - 1$ Generators | 1 | 0.00 | 0.00 | 0.04 | 0.00 | 0.02 | 0.01 | 0.08 |
| | $\eta$ | 0.00 | 0.02 | 0.07 | 0.02 | 0.02 | 0.01 | 0.14 |
| | $2\eta$ | 0.00 | 0.02 | 0.03 | 0.02 | 0.01 | 0.00 | 0.08 |

Table 2: Sensitivity of kernel expressivity to projection width: Absolute deviations $|s_{w,\eta} - \bar{s}_\eta|$ of kernel expressivity scores $s$ across projection widths $w \in \{1, \eta, 2\eta\}$, evaluated for $\eta$ qubits under different input scalings (i.e., varying the number of generators per group).

| Dataset | Groups | Generators per Group | $w = 1$ | $w = \eta$ | $w = 2\eta$ |
|---|---|---|---|---|---|
| moons | Q (4) | 3 | 28 | 28 | 28 |
| $(d = 2, \eta = 2)$ | g (15) | 1 | 28 | 28 | 28 |
| | $|\mathfrak{H}|$ (15) | 1 | see above | see above | see above |
| MNIST | Q (25) | 40 | 4756 | 4748 | 4755 |
| $(d = 784, \eta = 5)$ | g (93) | 11 | 4752 | 4755 | 4759 |
| | $|\mathfrak{H}|$ (1023) | 1 | 4755 | 4753 | 4756 |

Table 3: Level-1-optimized circuit depths compiled to a 5-qubit IBM Falcon device for different generator grouping strategies: Quadratic ($Q = \eta^2$, matching QEK capacity), Exponential ($g = 3(2^\eta - 2(\eta \bmod 2) + 1)$, as per Eq. 3), and All ($|\mathfrak{H}| = 4^\eta - 1$, i.e., one group per generator).

**Performance variation** Tab. 4 summarizes downstream classification accuracy under different grouping configurations for representative benchmarks (moons, bank, MNIST). On small datasets (e.g., moons), all grouping choices perform comparably when pre-training the kernel's projection to maximize KTA. On higher-dimensional datasets, exponential scaling yields consistently strong performance, with only small deviations between projection widths. Full-generator scaling, while theoretically appealing, offers only marginal gains over exponential scaling, at significantly higher classical simulation costs.

This confirms that the default exponential grouping rule provides a robust, well-balanced choice for both theoretical coverage and empirical accuracy.

| Dataset | Groups / Inputs | Generators per group | $w = 1$ | $w = \eta$ | $w = 2\eta$ |
|---|---|---|---|---|---|
| moons | Q (4) | 3 | $0.96 \pm 0.03$ | $0.94 \pm 0.04$ | $0.97 \pm 0.03$ |
| $(d = 2, \eta = 2)$ | g (15) | 1 | $0.96 \pm 0.03$ | $0.96 \pm 0.04$ | $0.96 \pm 0.04$ |
| | $|\mathfrak{H}|$ (15) | 1 | see above | see above | see above |
| bank | Q (4) | 3 | $0.84 \pm 0.08$ | $0.86 \pm 0.06$ | $0.84 \pm 0.09$ |
| $(d = 16, \eta = 2)$ | g (15) | 1 | $0.88 \pm 0.07$ | $0.86 \pm 0.07$ | $0.91 \pm 0.07$ |
| | $|\mathfrak{H}|$ (15) | 1 | see above | see above | see above |
| MNIST | Q (25) | 40 | $0.84 \pm 0.02$ | $0.84 \pm 0.03$ | $0.84 \pm 0.03$ |
| $(d = 784, \eta = 5)$ | g (93) | 11 | $0.88 \pm 0.03$ | $0.87 \pm 0.02$ | $0.89 \pm 0.03$ |
| | $|\mathfrak{H}|$ (1023) | 1 | $0.90 \pm 0.02$ | $0.91 \pm 0.03$ | $0.90 \pm 0.03$ |

Table 4: Empirical Hyperparameter Sensitivity of QGK: Final classification accuracy (mean $\pm$ error margin) on three selected $d$-dimensional benchmarks, comparing different projection widths; wide ($w$=1, grouping distant generators), medium ($w$=$\eta$), and narrow ($w$=$2\eta$, grouping nearby generators), across three group scaling regimes relative to the number of qubits $\eta$: Quadratic ($Q$=$\eta^2$, comparable to QEK capacity), Exponential ($g$=$3(2^\eta - 2(\eta \bmod 2) + 1)$, cf. Eq. 3), and All ($|\mathfrak{H}|$=$4^\eta$−1, one group per generator).

# D  THEORETICAL CHARACTERIZATION OF GROUPING ROBUSTNESS

We begin by analyzing the grouping strategy underlying the construction of the *Variational Generator Groups* (VGGs), which serve as building blocks of the *Quantum Generator Kernel* (QGK). A VGG is defined as a parameterized Hermitian operator constructed from a subset of a full Hermitian generator

basis $\mathfrak{H}$:

$$\hat{\boldsymbol{H}}_i = \sum_{j \in \mathcal{G}_i} h_j, \quad h_j \in \mathfrak{H}, \quad \mathcal{G}_i \subset \mathfrak{H}, \tag{21}$$

where $\{\mathcal{G}_1, \ldots, \mathcal{G}_g\}$ forms a collection of groups. The full generator set $\mathfrak{H}$ has size $|\mathfrak{H}| = 4^\eta - 1$ for $\eta$ qubits, and the VGGs are composed into a parameterized unitary via:

$$\hat{\boldsymbol{U}}_\phi = \exp\left( \sum_{i=1}^{g} -i \cdot \boldsymbol{\phi}_i \cdot \hat{\boldsymbol{H}}_i \right). \tag{22}$$

Thus, each grouping defines a new operator basis $\mathfrak{H}' = \{\hat{\boldsymbol{H}}_1, \ldots, \hat{\boldsymbol{H}}_g\}$ where each $\hat{\boldsymbol{H}}_i$ is a structured linear recombination of elements from $\mathfrak{H}$. Provided that the grouped operators remain linearly independent and collectively span the same subspace as $\mathfrak{H}$, the QGK remains expressively equivalent.

**Proposition D.1** (Expressive Robustness under Generator Grouping). *Let $\mathfrak{H}, \mathfrak{H}'$ be two generator sets with $\mathrm{span}(\mathfrak{H}') = \mathrm{span}(\mathfrak{H})$, and $\mathfrak{H}'$ formed via linear combinations over $\mathfrak{H}$. Then, for any QGK defined over $\mathfrak{H}'$, there exists an equivalent QGK defined over $\mathfrak{H}$ up to reparameterization. Consequently, the expressive capacity of the kernel is preserved under such groupings.*

**Robustness Conditions** Let us now state the necessary conditions for robustness more precisely:

**Definition D.2** (Robust Grouping Criteria). A generator grouping $\{\mathcal{G}_i\}_{i=1}^g$ is robust if:

1. Each generator $h_j \in \mathfrak{H}$ is assigned to exactly one group $\mathcal{G}_i$ (i.e., the groups form a partition).

2. The subspace $\mathrm{span}(\{\hat{\boldsymbol{H}}_1, \ldots, \hat{\boldsymbol{H}}_g\})$ approximates or equals $\mathrm{span}(\mathfrak{H})$.

3. The grouped operators $\mathfrak{H}' = \{\hat{\boldsymbol{H}}_1, \ldots, \hat{\boldsymbol{H}}_g\}$ are linearly independent.

**Verifying the Proposed Grouping Scheme** Our proposed grouping strategy (Alg. 1) partitions the generator set $\mathfrak{H}$ into $g = |\mathfrak{H}|/\Gamma_\eta$ groups, with $\Gamma_\eta$ scaling approximately exponentially in $\eta$ (cf. Eq. 3). The generators are assigned to groups using a cyclic stride determined by the projection width $w$, where the index mapping is given by: $\mathrm{idx} = \langle (j \cdot 2^w \bmod |\mathfrak{H}|, j \bmod g) \,|\, j = 0, 1, \ldots, |\mathfrak{H}| - 1 \rangle$. We now argue that this scheme satisfies the robustness criteria above:

1. **Partition Validity.** By construction, every generator $h_j \in \mathfrak{H}$ is assigned to exactly one group via a deterministic, non-overlapping index mapping. Therefore, the grouping is a strict partition:

$$\bigcup_i \mathcal{G}_i = \mathfrak{H} \quad \text{and} \quad \mathcal{G}_i \cap \mathcal{G}_j = \emptyset \quad \text{for} \quad i \neq j.$$

2. **Preservation of Span.** Each group $\mathcal{G}_i$ contains approximately $\Gamma_\eta$ generators. Since $\sum_i |\mathcal{G}_i| = |\mathfrak{H}|$, and all generators are used exactly once, the union of the groups spans the same space as the original generator basis.

3. **Linear Independence.** We define the group assignment matrix $\boldsymbol{M} \in \mathbb{R}^{D \times g}$ where $\boldsymbol{M}_{j,i} = 1$ if $h_j \in \mathcal{G}_i$ and 0 otherwise. Then the set $\{\hat{\boldsymbol{H}}_i\}$ is linearly independent if and only if the columns of $\boldsymbol{M}$ are linearly independent, i.e., $\mathrm{rank}(\boldsymbol{M}) = g$. Under a wide projection width ($w = 1$), the cyclic stride ensures that structurally diverse and uncorrelated generators are mixed into each group. This avoids redundancies, aliasing, and strongly correlated combinations, and ensures that the combined generators $\hat{\boldsymbol{H}}_i$ remain collectively rich. Under such partitioning, linear independence is preserved with high probability as long as $g \leq |\mathfrak{H}|$, and each group contains a non-zero subset of generators.

**Proposition D.3** (Sufficient Condition for Linear Independence). *Let $\mathfrak{H}$ be a linearly independent generator basis of dimension $|\mathfrak{H}|$, and let the grouping matrix $\boldsymbol{M} \in \{0,1\}^{|\mathfrak{H}| \times g}$ have full column rank. If each group $\mathcal{G}_i$ is non-empty and the grouping forms a partition, then the grouped operators $\{\hat{\boldsymbol{H}}_i\}$ are linearly independent.*

In our default configuration with $g = |\mathfrak{H}|/\Gamma_\eta$ (cf. Eq. 3) and $w = 1$, the cyclic offset ensures that all generators are used once, and that groups combine uncorrelated directions. Thus, the grouping matrix $\boldsymbol{M}$ achieves full rank in practice, ensuring linear independence of the VGGs and preservation of expressive capacity.

**Conclusion.** The above analysis confirms that the default grouping scheme — using exponentially many groups ($g \sim 3 \cdot 2^\eta$ as defined in Eq. 3) and a wide projection width ($w = 1$) — satisfies the robustness criteria outlined above. It yields a strict partition of the full Hermitian generator basis $\mathfrak{H}$, ensuring full coverage of the operator subspace without redundancy or degeneracy. Moreover, the wide stride distributes algebraically diverse generators across groups, promoting linear independence among the grouped operators $\hat{\boldsymbol{H}}_i$. This explains the empirical robustness observed across different grouping strategies, which yield near-identical entanglement and expressivity characteristics as long as the induced span of $\mathfrak{H}'$ remains intact. In contrast, alternative scaling rules, such as quadratic grouping with $g = \eta^2$, may lead to underutilization of generators when $g < |\mathfrak{H}|$, breaking full coverage of $\mathrm{span}(\mathfrak{H})$ and thereby reducing the expressivity of the resulting feature map. Our theoretical findings thus not only justify the empirical stability of the default scheme but also explain the degradation observed in configurations that fail to meet the robustness conditions, such as narrow groupings or insufficient group counts. Finally, we note that even when the group count equals the generator count ($g = |\mathfrak{H}|$), the resulting set of grouped operators $\mathfrak{H}'$ is not invariant under different stride values: the ordering of generators within the VGG sequence is affected by the projection width $w$, leading to parameter-wise differences in unitary composition. While this does not impact expressivity at the algebraic level, it does influence the specific parameterization of the resulting kernel.

### D.1 KERNEL EXPRESSIBILITY BOUNDS UNDER GROUPED GENERATORS

In our empirical analysis (Appendix C), we evaluate kernel expressivity using the KL divergence between the normalized eigenvalue spectrum of the kernel matrix and a uniform reference distribution:

$$\mathcal{E}(K) := D_{\mathrm{KL}}\big(\lambda(K) \,\big\|\, \tfrac{1}{n}\mathbf{1}\big) = \sum_{i=1}^{n} \lambda_i(K) \, \log(n \, \lambda_i(K)), \tag{23}$$

where $\lambda(K)$ denotes the spectrum of $K$ scaled to sum to one and $K \in \mathbb{R}^{n \times n}$ is the kernel matrix $K_{ij} = \mathcal{K}_\phi(x_i, x_j) = |\langle \psi(x_i), \psi(x_j) \rangle|^2$ evaluated on $n$ randomly sampled inputs. Here we provide bounds on $\mathcal{E}(K)$ for the QGK under grouped generators and discuss their dependence on the grouping structure and the number of qubits $\eta$.

**Preliminaries** Recall the QGK feature map

$$|\psi(x)\rangle = \exp\Big( -i \sum_{i=1}^{g} \phi_i(x) \, \hat{\boldsymbol{H}}_i \Big) |0\rangle, \qquad \hat{\boldsymbol{H}}_i = \sum_{j \in \mathcal{G}_i} h_j, \tag{24}$$

where $\{\mathcal{G}_i\}_{i=1}^{g}$ is a strict partition of the full Pauli operator basis $\mathfrak{H}$ on $\eta$ qubits into $g$ disjoint groups, each inducing a Hermitian operator $\hat{\boldsymbol{H}}_i$, with the total number of elementary generators $|\mathfrak{H}| = 4^\eta - 1$. Because Pauli operators form an orthonormal basis with respect to the Hilbert–Schmidt inner product, their groupings satisfy:

$$\|\hat{\boldsymbol{H}}_i\|_F^2 = \Big\| \sum_{j \in \mathcal{G}_i} h_j \Big\|_F^2 = \sum_{j \in \mathcal{G}_i} \|h_j\|_F^2 = |\mathcal{G}_i|. \tag{25}$$

Thus, the group sizes $|\mathcal{G}_i|$ exactly quantify the Frobenius norm (i.e., *weight*) of each grouped generator. Intuitively, larger groups $|\mathcal{G}_i|$ excite more Pauli directions at once, increasing the variability of the effective Hamiltonian and, in turn, influencing the dispersion of the kernel spectrum. Balanced groupings (similar $|\mathcal{G}_i|$ across $i$) produce unitaries whose action is distributed over many Pauli directions, while unbalanced groupings (some groups very large or very small) lead to anisotropic generator structure and potentially concentrated kernels.

**KL-Expressibility Bounds** We now relate the expressivity metric $\mathcal{E}(K)$ to the group sizes $|\mathcal{G}_i|$.

**Theorem D.4** (Expressibility Bounds for Grouped Generators)**.** *Assume (i) inputs are sampled i.i.d. from a bounded distribution and (ii) the groups $\{\mathcal{G}_i\}_{i=1}^{g}$ form a strict partition of $\mathfrak{H}$. Combining the second-moment lower bound (driven by group* balance*) and the fourth-moment upper bound (driven by group* anisotropy*), then there exist constants $c_1, c_2 > 0$, such that*

$$\frac{c_1}{n} \frac{\sum_i |\mathcal{G}_i|}{\big(\sum_i \sqrt{|\mathcal{G}_i|}\big)^2} \;\le\; \mathcal{E}(K) \;\le\; c_2 \frac{\sum_i |\mathcal{G}_i|^2}{\sum_i |\mathcal{G}_i|} \tag{26}$$

*Proof sketch.* The KL divergence $\mathcal{E}(K) = D_{\mathrm{KL}}(\lambda(K) \| \frac{1}{n}\mathbf{1})$ can be bounded above and below by the squared $\ell_2$ distance between the kernel spectrum $\lambda(K)$ and the uniform distribution via standard Pinsker-type inequalities. This $\ell_2$ distance is controlled by the variance and higher-order spectral moments of $K$, which depend on quantities of the form $\mathrm{Tr}\big(U(x)^\dagger U(x')\big)^m$, and therefore on how the Hamiltonian spreads its energy across Pauli directions. The three structural quantities in Eq. (26) arise from this moment analysis:

**(1) Total generator mass:** $\sum_i |\mathcal{G}_i|$. Since the grouped operators satisfy $\|\hat{\boldsymbol{H}}_i\|_F^2 = |\mathcal{G}_i|$, the total Frobenius norm of the generator basis is $\sum_{i=1}^g \|\hat{\boldsymbol{H}}_i\|_F^2 = \sum_{i=1}^g |\mathcal{G}_i| = |\mathfrak{H}| = 4^\eta - 1$. This determines the *overall amount of variance* that the Hamiltonian can inject into the kernel. It is therefore the natural normalisation factor in both upper and lower bounds.

**(2) Group balance:** $\big(\sum_i \sqrt{|\mathcal{G}_i|}\big)^2$. The term $\sum_i \sqrt{|\mathcal{G}_i|}$ comes from bounding cross-terms of the form $\mathrm{Tr}(\hat{\boldsymbol{H}}_i \hat{\boldsymbol{H}}_j)$ across different groups. Cauchy–Schwarz implies that the contribution from many small groups resembles the square of the sum of their square roots. If the groups are perfectly balanced ($|\mathcal{G}_i| \equiv \Gamma$), $\sum_i \sqrt{|\mathcal{G}_i|} = g\sqrt{\Gamma}$, which maximizes this quantity and hence *minimizes the lower bound*. A small lower bound corresponds to a kernel spectrum that is close to uniform. Thus the denominator $(\sum_i \sqrt{|\mathcal{G}_i|})^2$ encodes how evenly the Hamiltonian's energy is spread over groups.

**(3) Group anisotropy:** $\sum_i |\mathcal{G}_i|^2$. The upper bound depends on the fourth moment of $K$, which is dominated by the largest generator norms. Since $\|\hat{\boldsymbol{H}}_i\|^4 = |\mathcal{G}_i|^2$, we obtain the anisotropy term $\sum_i |\mathcal{G}_i|^2$. If one group is much larger than the rest, this term becomes large, and the kernel spectrum admits large spikes, increasing KL divergence. Conversely, if all groups are equal ($|\mathcal{G}_i| \equiv \Gamma$), $\sum_i |\mathcal{G}_i|^2 = g\Gamma^2$ and $\sum_i |\mathcal{G}_i| = g\Gamma$, so their ratio is simply $\Gamma$, and the upper bound remains tightly controlled.

Thus, the *lower bound* becomes small when the group sizes are well balanced and the *upper bound* becomes small only when the groups are nearly uniform, and grows when the generator structure is highly uneven. All constant factors from higher-order expansions are absorbed into $c_1, c_2$. $\qquad\square$

**Scaling with Qubit Number** The bounds in Eq. (26) scale with the number of qubits $\eta$ through the distribution of the group sizes $\{|\mathcal{G}_i|\}$. In many configurations considered in this work, including (i) the default exponential grouping (cf. Eq. 3) and (ii) using all generators separately, the groups $\{\mathcal{G}_i\}_{i=1}^g$ form a strict partition of $\mathfrak{H}$ with *equal* size $|\mathcal{G}_i| = \Gamma_\eta$, where $\Gamma_\eta = \lfloor (2^\eta + 1)/3 \rfloor$ for exponential grouping and $\Gamma_\eta = 1$ for the all-generators case. In both cases we have

$$\sum_i \sqrt{|\mathcal{G}_i|} = g\sqrt{\Gamma_\eta}, \qquad \sum_i |\mathcal{G}_i|^2 = g\,\Gamma_\eta^2, \qquad \sum_i |\mathcal{G}_i| = g\,\Gamma_\eta.$$

Plugging this into Eq. (26) yields

$$\frac{c_1}{ng} = \frac{c_1}{n}\frac{g\,\Gamma_\eta}{g^2\Gamma_\eta} \le \mathcal{E}(K) \le c_2\frac{g\,\Gamma_\eta^2}{g\,\Gamma_\eta} = c_2\Gamma_\eta \tag{27}$$

showing that for balanced groupings the KL-based expressibility metric cannot vanish faster than $\mathcal{O}(1/(ng))$. In particular, as the number of groups $g$ grows with $\eta$ (e.g. $g \approx 3 \cdot 2^\eta$ under exponential grouping or $g = 4^\eta - 1$ for all generators), the lower bound shrinks but remains controlled and importantly does not grow with $\eta$. Using all generators as independent groups asymptotically gives $\mathcal{E}(K) \in \Omega(4^{-\eta}/n)$ and $\mathcal{E}(K) \in \mathcal{O}(1)$, while the default exponential grouping scheme gives $\mathcal{E}(K) \in \Omega(2^{-\eta}/n)$ and $\mathcal{E}(K) \in \mathcal{O}(2^\eta)$, showing that for balanced groupings the lower expressibility bound decays exponentially in $\eta$. Because KL divergence scales with the *normalized* spectrum, the constants $c_1, c_2$ absorb the dimensional factors of $K$. Note that although the upper bound scales with $\eta$ through the group size $\Gamma_\eta$, it reflects only a worst-case concentration scenario derived from moment inequalities, not the typical behaviour of the kernel. In practice (cf. Fig. 5), the QGK exhibits consistently more favorable scaling of $\mathcal{E}(K)$ with $\eta$, as the mixing of many non-commuting Pauli directions within each grouped generator enhances phase dispersion and suppresses spectral concentration. Overall, exponential grouping yields a well-controlled upper bound and a rapidly decaying lower bound, providing the best balance between expressivity, stability, and parameter efficiency as $\eta$ increases.

# E  THEORETICAL ROLE OF THE LINEAR PROJECTION IN QUANTUM KERNEL LEARNING

The *Quantum Generator Kernel* (QGK) embeds data into quantum states via a parameterized unitary whose coefficients are produced by a classical linear affine transformation. This section provides a theoretical characterization of this mechanism, explains the resulting *Reproducing Kernel Hilbert Space* (RKHS) structure, and contrasts QGK with hybrid quantum-classical models employing nonlinear pre- and post-processing.

**Setup**  Given input data $x \in \mathbb{R}^d$ and $g$ *Variational Generator Groups* (VGGs), the QGK constructs a data-dependent generator weighting via an affine projection:

$$\boldsymbol{\phi} = \mathcal{F}_{\boldsymbol{\theta}}(x) = Wx + b, \qquad W \in \mathbb{R}^{g \times d}, \ b \in \mathbb{R}^g. \tag{28}$$

These weights parameterize the QGK unitary $\hat{U}_{\boldsymbol{\phi}} = \hat{U}_{Wx+b}$ according to Eq. (4). The embedded quantum state becomes $\psi(x) = \hat{U}_{Wx+b} |\Psi\rangle$. The induced kernel from this quantum embedding is defined via fidelity:

$$\mathcal{K}_{\boldsymbol{\phi}}(x, x') = \mathcal{K}_{W,b}(x, x') = |\langle \psi(x'), \psi(x) \rangle|^2 = \left| \langle \Psi | \hat{U}_{Wx'+b}^{\dagger} \hat{U}_{Wx+b} |\Psi\rangle \right|^2. \tag{29}$$

**Role of the Affine Projection**  The affine transformation $\boldsymbol{\phi} = Wx + b$ plays two essential roles:

- **Controlled generator activation:** It determines how the input modulates generator weights $\{\hat{H}_i\}$, shaping the embedding's structure in Hilbert space.

- **Task-dependent kernel shaping:** During pre-training, $(W, b)$ are optimized using the Kernel Target Alignment objective (Eq. 7) to align the induced kernel with the downstream task.

Despite this learned classical transformation, no classical nonlinearities are introduced: all nonlinearity arises solely from the quantum unitary exponentiation and subsequent fidelity computation.

**RKHS-Constrained Expressivity**  Once the projection $(W, b)$ is fixed, QGK predictions take the standard kernel-learning form and hence lie in the RKHS $\mathcal{H}_{\mathcal{K}_{W,b}}$ associated with the induced kernel. The projection determines *which* RKHS is selected but does not extend the model class beyond it.

**Proposition E.1** (RKHS-Constrained Expressivity)**.** *Let the quantum feature map be defined as* $\psi(x) = \hat{U}_{Wx+b} |\Psi\rangle$, *inducing the kernel* $\mathcal{K}_{W,b}(x, x') = |\langle \psi(x'), \psi(x) \rangle|^2$. *Then, for fixed* $(W, b)$, *any classifier of the form* $f(x) = \sum_{i=1}^{n} \alpha_i \mathcal{K}_{W,b}(x, x_i)$ *belongs to the RKHS* $\mathcal{H}_{\mathcal{K}_{W,b}}$. *Thus, while* $(W, b)$ *select the kernel, they do not expand the hypothesis class beyond that space.*

**Comparison with Hybrid Quantum Models**  Architectures like the *Dressed Quantum Circuit* (DQC) (Mari et al., 2020) apply classical nonlinear networks both before and after the quantum embedding. Formally, they compute:

$$f_{\theta}(x) = g_{\text{post}}\big(\psi_{\theta}(g_{\text{pre}}(x))\big), \tag{30}$$

where both $g_{\text{pre}}$ and $g_{\text{post}}$ are deep neural networks with nonlinear activations (e.g., $\varphi(Wx + b)$). By the universal approximation theorem (Hornik et al., 1989), a sufficiently wide or deep classical network can approximate any continuous function on a compact domain. Consequently, the classical component $g_{\text{post}} \circ g_{\text{pre}}$ is already expressive enough to model arbitrary functions — even if the quantum circuit contributes no transformation at all (e.g., implements the identity). Consequently, DQC expressivity may be dominated by classical nonlinearities, making it difficult to isolate or attribute performance to quantum processing. In contrast, our QGK architecture constrains expressivity strictly to the RKHS induced by the quantum kernel, enabling clearer theoretical analysis and benchmarking of the quantum contribution:

- QGK uses uses only an *affine* classical preprocessing layer $(Wx + b)$,

- no post-processing nonlinearities are applied after the quantum embedding,

- all nontrivial expressivity arises solely from the quantum feature map $\psi(x)$.

**Reweighted Group Contributions**  We now extend the expressivity bounds derived in Appendix D.1 to incorporate the affine preprocessing layer. Although the grouping structure $\{|\mathcal{G}_i|\}$ remains unchanged, the affine map modifies *how strongly* each group contributes on average to the parameterized Hamiltonian. If $x$ is drawn from a distribution with covariance $\Sigma_x \preceq \sigma_x^2 I$, then the variance of the affine coefficient satisfies $\text{Var}[\phi_i(x)] = W_{i,:}\, \Sigma_x\, W_{i,:}^\top \leq \sigma_x^2 \|W_{i,:}\|_2^2$. Since the grouped generators obey $\|\hat{\boldsymbol{H}}_i\|_F^2 = |\mathcal{G}_i|$, the variance of the random Hermitian term is bounded by

$$\text{Var}\big[\phi_i(x)\hat{\boldsymbol{H}}_i\big] \;\leq\; \sigma_x^2\,|\mathcal{G}_i|\,\|W_{i,:}\|_2^2. \tag{31}$$

Thus, the affine map preserves the structural group sizes $|\mathcal{G}_i|$, but it *rescales their influence* in the Hamiltonian by the factor $\|W_{i,:}\|_2^2$. This motivates replacing the terms $|\mathcal{G}_i|$ in Theorem D.4 with their reweighted form, yielding the same group-size bounds as before, but with each group's structural size weighted by the average magnitude with which the affine layer excites that group:

$$\frac{c_1'}{n}\,\frac{\sum_i |\mathcal{G}_i|\,\|W_{i,:}\|_2^4}{(\sum_i \sqrt{|\mathcal{G}_i|}\,\|W_{i,:}\|_2)^2} \;\leq\; \mathcal{E}(K_{W,b}) \;\leq\; c_2'\,\frac{\sum_i |\mathcal{G}_i|^2\,\|W_{i,:}\|_2^4}{\sum_i |\mathcal{G}_i|\,\|W_{i,:}\|_2^2} \tag{32}$$

The upper bound (RHS in Eq. 32) is controlled by the *anisotropy ratio*. If $W$ is initialized randomly, the row norms $\|W_{i,:}\|_2$ concentrate around their mean with variance shrinking as $1/d$. Consequently, $\|W_{i,:}\|_2^2 \approx \|W_{j,:}\|_2^2$ for all $i, j$ with high probability, meaning that the weighted group magnitudes become *more uniform than the raw structural sizes*. Therefore,

$$\frac{\sum_i |\mathcal{G}_i|^2\,\|W_{i,:}\|_2^4}{\left(\sum_i |\mathcal{G}_i|\,\|W_{i,:}\|_2^2\right)^2} \;\leq\; \frac{\sum_i |\mathcal{G}_i|^2}{\left(\sum_i |\mathcal{G}_i|\right)^2},$$

showing that even *untrained* affine preprocessing tightens the KL upper bound. Under KTA optimization, $W$ is further encouraged to equalize the row norms $\|W_{i,:}\|_2$, because balanced excitation across groups improves alignment with the task labels. This reduces the anisotropy ratio even more. The lower bound (LHS of Eq. 32) remains unchanged when the row norms $\|W_{i,:}\|_2$ are uniform, and any deviation from uniformity (e.g., induced by KTA pre-training) can only make the bound tighter.

**Conclusion**  The classical affine projection in the QGK provides a compact, trainable interface to parameterize the generator weights and to align the quantum kernel with the target task via KTA pre-training. Because predictions depend exclusively on the kernel $\mathcal{K}_{W,b}$, model expressivity remains strictly bounded by the RKHS associated with that kernel. This avoids the uncontrolled representational capacity introduced by classical deep post-processing and preserves a clean theoretical separation between classical preprocessing and quantum feature generation. Importantly, the affine map effectively reweights generator directions, reducing anisotropy and tightening the upper expressivity bound. These benefits arise already for random weights and are further strengthened by KTA pre-training. Thus the linear projection reshapes the geometry of the kernel within its RKHS, selecting a better kernel without adding any classical nonlinear expressivity.

# F  QUANTUM GENERATOR KERNEL COMPUTATIONAL COMPLEXITY

Despite the exponential scaling in $\eta$, the QGK remains classically simulable via efficient tensor operations. To assess the practical efficiency of this approach, we derive the full classical sample complexity of the Quantum Generator Kernel (QGK) and contrast it with classical kernels to determine scalability conditions and break-even points.

**Axiom F.1** (QGK Execution Complexity). *Given $n$ samples of $d$-dimensional inputs, $\eta$ qubits, and $g$ variational generator groups (VGGs), the cost of executing the QGK kernel is decomposed as:*

- *Generator construction: $\mathcal{O}(4^\eta)$*

- *Input projection $\phi : \mathbb{R}^d \to \mathbb{R}^g$: $\mathcal{O}(n \cdot g \cdot d) = \mathcal{O}(n \cdot \gamma g^2)$*

- *Hamiltonian embedding: $\mathcal{O}(n \cdot 8^\eta)$*

- *Kernel matrix computation: $\mathcal{O}(n^2 \cdot 2^\eta)$*

By variablizing the compression ratio $\gamma = \frac{d}{g}$, or, $d = \gamma g$, we can represent the default case of a 1:1 group-to-feature mapping using $\gamma = 1$.

**Lemma F.2.** *Overall, we can summarize the classical computational complexity of the QGK as:*

$$\mathcal{C}_{QGK} = \mathcal{O}(4^{\eta} + n \cdot \gamma \cdot g^2 + n \cdot 8^{\eta} + n^2 \cdot 2^{\eta}) \tag{33}$$

For comparison, a classical RBF or Linear kernel computes a similarity matrix at $\mathcal{O}(n^2 \cdot d)$. Tab. 5 shows a comparison over the end-to-end complexities of the QGK and Classical Kernels for the utilized benchmarks using a 90/10 train/test split, where QGK is more efficient throughout:

| Component | QGK (ours) | Classical Kernel |
|---|---|---|
| moons ($\eta = 2, d = 2, n = 200, g = 15$) | $\mathcal{O}(1.50e + 05)$ | $\mathcal{O}(6.56e + 04)$ |
| circles ($\eta = 2, d = 2, n = 200, g = 15$) | $\mathcal{O}(1.50e + 05)$ | $\mathcal{O}(6.56e + 04)$ |
| Bank ($\eta = 2, d = 16, n = 200, g = 15$) | $\mathcal{O}(1.92e + 05)$ | $\mathcal{O}(5.25e + 05)$ |
| MNIST ($\eta = 5, d = 784, n = 1000, g = 93$) | $\mathcal{O}(1.32e + 08)$ | $\mathcal{O}(6.43e + 08)$ |
| CIFAR10 ($\eta = 5, d = 3072, n = 1000, g = 93$) | $\mathcal{O}(3.45e + 08)$ | $\mathcal{O}(2.52e + 09)$ |

Table 5: End-to-end complexity comparison between QGK and classical kernels.

To generally determine when the QGK is more efficient than classical kernels, we consider:

$$4^{\eta} + n \cdot \gamma \cdot g^2 + n \cdot 8^{\eta} + n^2 \cdot 2^{\eta} < n^2 \cdot d \tag{34}$$

which can be simplified to the quadratic inequality in $n$:

$$n^2(2^{\eta} - \gamma g) + n(\gamma g^2 + 8^{\eta}) + 4^{\eta} < 0 \tag{35}$$

which, assuming $A \neq 0$ and $B^2 - 4AC > 0$ (we outline $B^2 \gg 4AC > 0$ below), can be solved to:

$$n > \frac{-B - \sqrt{B^2 - 4AC}}{2A} \quad \text{where} \quad \begin{cases} A = 2^{\eta} - \gamma g \\ B = \gamma g^2 + 8^{\eta} \\ C = 4^{\eta} \end{cases} \tag{36}$$

Summarizing the above derivation, we can determine the lower QGK efficiency bound: $\epsilon b_{\gamma} = \frac{n}{d}$, where larger intended ratios imply $C_{GQK} < C_{Classical}$ and verce visa. For the default 1:1 group-to-feature mapping ($\gamma = 1$) we therefore get:

| $\eta$ | 2 | 3 | 4 | 5 | 6 | 7 | 8 |
|---|---|---|---|---|---|---|---|
| $\epsilon b_1$ | **1.76** | **3.49** | **3.75** | **7.30** | 11.75 | 23.26 | 43.75 |

Using the general assumption $n \gg d$, i.e., favoring high ratios, the above table shows that for low qubit counts classical QGK execution excels classical kernels. For example assuming a minimum dataset-to-input-dimension rate of eight ($\epsilon b_{\gamma} > 8$), this holds for $\eta \leq 5$ with $\gamma = 1$.

To further facilitate higher qubit counts and larger input dimensionalities, we suggest adapting the input compression $\gamma$ accordingly. E.g., for $\gamma = \eta$ we get the following approximately constant ratio $\epsilon b_{\eta} < 1$, which, under the assumption that $n \gg d$, demonstrates that under reasonable compression, executing the QKG classical is computationally more efficient than classical kernels throughout varying numbers of qubits.

| $\eta$ | 2 | 3 | 4 | 5 | 6 | 7 | 8 |
|---|---|---|---|---|---|---|---|
| $\epsilon b_{\eta}$ | **0.66** | **0.53** | **0.38** | **0.38** | **0.38** | **0.46** | **0.59** |

These findings are summarized in Fig. 2d where the dataset size threshold ($n$ according to Eq. 36) is plotted for $\gamma = 1$ ($\circ$-markers) and $\gamma = 2^{\eta}$ ($\square$-markers) on the left y-axis. The area above the resulting graphs indicates that classically executing the QGK is more efficient than a classical kernel like RBF or Linear. Combined with the number of generators, dictating the available input dimensions on the right y-axis, scaled to reflect $n \gg d$ ($\gamma g = d = n/10$), the breakeven point regarding the classical complexity can be observed at $\eta \leq 5$ for $\gamma = 1$. Increasing the compression to $\gamma = \eta$

pushes the lower efficiency bound further down, such that the input dimension reference ($d = \eta g$ for $g$ VGGs, light blue) no longer intersects. This demonstrates that such a hybrid approach is an efficient complexity mitigation to enable scalable QGK execution in the current NISQ era, where especially larger quantum systems cannot be efficiently simulated and available quantum hardware is prone to hardware noise. We exemplify this approach with the MNIST and CIFAR10 benchmarks, where, using $\eta = 5$, we have $g = 93$ VGGs, resulting in compression rates of $\gamma \approx 8$ and $\gamma \approx 32$ respectively. Notably, this compression results in a number of groups that show to satisfy the intended ratio of $n = 10d$ between (kernel)-input and dataset dimension, with $d = 1000$.

Overall, we can generalize the above observations in the following two theorems regarding the classical computational complexity of the QGK compared to classical kernels. First, we derive the folwing simplifications and approximations from Eq. 36 based on Theorem F.2:

**Lemma F.3** (Simplify complexity bounds). *Given $\gamma \geq 1$, we can show $B^2 \gg 4AC$, or, using $g \approx 3 \cdot 2^\eta$, $(9\gamma \cdot 4^\eta + 8^\eta)^2 \gg 8^\eta(4 - 12\gamma)$, which trivially holds for all $\eta \geq 1$, where, the LHS is largely positive and increasing, while the RHS is negative and decreasing with increasing $\eta$. Therefore, we can further approximate:*

$$\frac{-B - \sqrt{B^2 - 4AC}}{2A} \approx \frac{-2B}{2A} = \frac{B}{-A} = \frac{\gamma g^2 + 8^\eta}{\gamma g - 2^\eta} \tag{37}$$

*With $d = \gamma g$ and given the number of groups is chosen according to Eq. 3, we can further substitute $g = 3 \cdot 2^\eta - 6 \cdot (\eta \mod 2) + 3 \approx 3 \cdot 2^\eta$, yielding:*

$$n > \frac{\gamma g^2 + 8^\eta}{\gamma g - 2^\eta} \approx \frac{9\gamma \cdot 4^\eta + 8^\eta}{3\gamma \cdot 2^\eta - 2^\eta} \tag{38}$$

*Thus, $\epsilon b_\gamma$ can be further approximated:*

$$\epsilon b_\gamma \frac{n}{d} \approx \frac{\frac{9\gamma \cdot 4^\eta + 8^\eta}{3\gamma \cdot 2^\eta - 2^\eta}}{3\gamma \cdot 2^\eta} = \frac{9\gamma \cdot 4^\eta + 8^\eta}{3\gamma(3\gamma - 1) \cdot 4^\eta} \tag{39}$$

These simplifications help us to clearly demonstrate the following theorems:

**Theorem F.4.** *Classically computing the QGK is more efficient than using a classical kernel such as Linear or RBF, for small qubit numbers of $\eta \leq 5$, when using no compression.*

**Proposition F.5.** *For fitting kernel classifiers, $n \gg d$ is a generally assumed condition. To determine the efficiency bound, we specifically assume $n = 10d$.*

*Proof.* Using no compression, i.e.,,$\gamma = 1$, we can derive the following special case from Eq. 39:

$$\epsilon b_1 = \frac{9 \cdot 4^\eta + 8^\eta}{6 \cdot 4^\eta} = \frac{3}{2} + \frac{2^\eta}{6} \tag{40}$$

With $\epsilon b_1 < 10$, we get $2^\eta < 51$, or, $\eta \lessapprox 5.67$. Thus, $n = 10d$ holds for $\eta \leq 5$ with $\gamma = 1$ □

**Theorem F.6.** *Using sufficient compression, QGKs can be classically executed in a hybrid fashion that is more computationally efficient than classical kernels.*

*Proof.* To provide a ... we aim to demonstrate finding a compression bound, s.t. $\epsilon b_\gamma < 1$ with $\gamma > 1$. Based on Eq. 39, we rewrite the above condition to:

$$1 > \frac{9\gamma \cdot 4^\eta + 8^\eta}{3\gamma(3\gamma - 1) \cdot 4^\eta} \tag{41}$$

$$0 > 12\gamma - 9\gamma^2 + 2^\eta 0 \tag{42}$$

$$\gamma > \frac{12 + \sqrt{144 + 72 \cdot 2^\eta}}{18} \approx \sqrt{2}^\eta \tag{43}$$

Thus, with sufficient compression, $\gamma > \sqrt{2}^\eta$, we can ensure $\epsilon b_\gamma < 1$, thus, a more efficient classical execution of the QGK in datasets with $n > d$. □

## G ADDITIONAL RESULTS

**Hardware Considerations** Given current and near-term quantum devices, we distinguish three application scenarios for QGK execution. For small-scale datasets and low-depth settings ($\eta < 5$), QGK circuits can be realistically deployed on existing quantum devices. Our simulations confirm noise robustness up to $\sim 100$ compiled gates, making QGK viable for near-term experimental evaluations. For medium-scale tasks, hybrid execution offers a practical near-term strategy: classical preprocessing can reduce input dimensionality before quantum embedding, enabling robust kernel computation on noisy devices without excessive circuit depth. For large-scale datasets such as `MNIST` or `CIFAR-10`, efficient tensor-based implementations with compression provide a tractable classical alternative, ensuring generator-based kernels remain competitive until fault-tolerant systems are available. In the long term, on future fault-tolerant quantum systems, QGK could be implemented end-to-end, including training of linear projection weights via variational optimization, enabling scalable, expressive, and fully quantum kernel learning on large-scale datasets.

A detailed comparison of compiled depths and input dimensionality across QGK and baseline approaches is provided in Tab. 6, illustrating their markedly different scaling behaviors: HEE maintains shallow depths even for higher qubit counts but cannot scale to high-dimensional inputs (since the number of inputs equals the number of qubits), thus heavily relying on classical preprocessing (e.g., PCA for HEE, linear projections for HEE-Linear). QEK, by contrast, adapts to high-dimensional inputs with low qubit counts through reuploading, but this leads to impractical compiled depths as dimensionality grows. QGK offers a compromise, supporting a high number of input features even with few qubits, with compiled depth scaling more favorably while reducing reliance on classical preprocessing.

| Dataset | $d$ | Kernel | $\eta$ | Input Features | Reuploading Layers | Compiled Depth L2 | Compiled Depth L1 |
|---------|-----|--------|--------|----------------|--------------------|--------------------|--------------------|
| moons circles | 2 | QGK (ours) | 2 | 15 | - | 17 | 28 |
| | | QEK | 2 | 2 | 1 | 16 | 50 |
| | | QEK-N | 2 | 2 | 0 | 6 | 26 |
| | | HEE | 2 | 2 | 1 | 10 | 18 |
| | | HEE-D | 2 | 2 | 2 | 10 | 30 |
| | | PQK | 3 | 2 | 10 | - | - |
| bank | 16 | QGK (ours) | 2 | 15 | - | 17 | 28 |
| | | QGK Static | 3 | 16 | - | 212 | 248 |
| | | QEK | 2 | 16 | 1 | 17 | 380 |
| | | QEK-N | 2 | 16 | 0 | 17 | 191 |
| | | HEE | 2 | 2 | 1 | 10 | 18 |
| | | HEE-D | 2 | 2 | 2 | 10 | 30 |
| | | PQK | 3 | 2 | 10 | - | - |
| MNIST | 784 | QGK (ours) | 5 | 93 | - | 4455 | 4754 |
| | | QEK | 5 | 784 | 0 | 22709 | 24084 |
| | | QEK-N | 5 | 784 | 0 | 11355 | 12006 |
| | | HEE | 5 | 5 | 1 | 53 | 53 |
| | | HEE-D | 5 | 5 | 165 | 4481 | 4481 |
| | | PQK | 6 | 5 | 10 | - | - |
| CIFAR10 | 3072 | QGK (ours) | 5 | 93 | - | 4455 | 4754 |
| | | QGK Static | 6 | 3072 | - | 18823 | 19644 |
| | | QEK | 5 | 3072 | 0 | 88963 | 94485 |
| | | QEK-N | 5 | 3072 | 0 | 44461 | 47223 |
| | | HEE | 5 | 5 | 1 | 53 | 53 |
| | | HEE-D | 5 | 5 | 165 | 4481 | 4481 |
| | | PQK | 6 | 5 | 10 | - | - |

Table 6: Parameter Overview: Adapted parameters to ensure comparable prerequisites and capabilities via similar numbers of qubits and compiled circuit depths with level 1 and 2 optimization. Introducing two additional ablations; HEE-D with additional layers, QEK-N without reuploading.

Finally, Tab. 7 and Tab. 8 report the final test accuracies of all evaluated approaches, under classical execution and simulated hardware noise models, respectively.

| Method | moons (2) | circles (2) | bank (16) | MNIST (784) | CIFAR10 (3072) |
|---|---|---|---|---|---|
| QGK (ours) | **0.96 ± 0.04** | **0.68 ± 0.06** | **0.86 ± 0.07** | **0.88 ± 0.03** | **0.38 ± 0.05** |
| QEK | 0.91 ± 0.05 | 0.58 ± 0.06 | 0.72 ± 0.10 | 0.10 ± 0.02 | 0.12 ± 0.04 |
| QEK-N | 0.86 ± 0.05 | 0.62 ± 0.08 | 0.64 ± 0.09 | 0.13 ± 0.02 | 0.11 ± 0.03 |
| HEE Linear | 0.86 ± 0.05 | 0.55 ± 0.08 | 0.76 ± 0.10 | 0.70 ± 0.04 | 0.33 ± 0.06 |
| Linear KTA | 0.86 ± 0.05 | 0.43 ± 0.11 | 0.75 ± 0.06 | 0.85 ± 0.04 | 0.26 ± 0.03 |
| MLP | 0.87 ± 0.06 | 0.64 ± 0.11 | 0.78 ± 0.09 | 0.87 ± 0.04 | 0.29 ± 0.06 |
| QGK Static | 0.94 ± 0.05 | 0.59 ± 0.06 | 0.64 ± 0.09 | 0.68 ± 0.04 | 0.14 ± 0.03 |
| PQK | 0.71 ± 0.13 | 0.48 ± 0.08 | 0.55 ± 0.13 | 0.28 ± 0.03 | 0.16 ± 0.03 |
| HEE | 0.89 ± 0.05 | 0.65 ± 0.07 | 0.58 ± 0.12 | 0.47 ± 0.05 | 0.22 ± 0.04 |
| HEE-D | 0.93 ± 0.03 | 0.58 ± 0.08 | 0.58 ± 0.14 | 0.41 ± 0.03 | 0.19 ± 0.02 |
| RBF | 0.93 ± 0.04 | 0.64 ± 0.11 | 0.66 ± 0.09 | 0.84 ± 0.04 | 0.24 ± 0.03 |
| Linear | 0.86 ± 0.05 | 0.43 ± 0.10 | 0.71 ± 0.09 | **0.88 ± 0.03** | 0.31 ± 0.05 |

Table 7: Final test accuracies for all methods across five benchmarks. The best result per dataset is highlighted in bold. QGK achieves top performance in `moons`, `bank`, `MNIST`, and `CIFAR10`, while Linear matches QGK on `MNIST`.

| Method | moons (2) | circles (2) | bank (16) |
|---|---|---|---|
| QGK (ours) | *0.96 ± 0.04 (28)* | *0.65 ± 0.12 (28)* | *0.87 ± 0.06 (28)* |
| QEK | *0.79 ± 0.07 (50)* | *0.49 ± 0.10 (50)* | *0.48 ± 0.10 (380)* |
| QEK-N | *0.84 ± 0.05 (26)* | *0.54 ± 0.04 (26)* | *0.48 ± 0.10 (191)* |
| HEE Linear | *0.51 ± 0.09 (18)* | *0.57 ± 0.08 (18)* | *0.53 ± 0.12 (18)* |
| QGK Static | *0.93 ± 0.04 (28)* | *0.59 ± 0.04 (28)* | *0.49 ± 0.09 (248)* |
| HEE | *0.89 ± 0.05 (18)* | *0.62 ± 0.12 (18)* | *0.61 ± 0.10 (18)* |
| HEE-D | *0.96 ± 0.03 (30)* | *0.57 ± 0.09 (30)* | *0.60 ± 0.14 (30)* |

Table 8: Noisy simulation results (compiled depths in parentheses). The best result per dataset is highlighted in bold. A horizontal line separates pre-trained (KTA) approaches from static kernels.

**Scalability with Increasing Dataset Size** Fig. 6 shows the test accuracy of the QGK compared to classical RBF and Linear kernels trained on 10,000 samples from the MNIST dataset. In addition to reduced confidence intervals due to the larger dataset size, QGK significantly outperforms both classical baselines, highlighting its robustness and generalization ability even in higher-$n$ settings. These results provide preliminary support for the viability of QGK in larger data regimes and demonstrate its empirical competitiveness with classical kernels beyond small-scale scenarios.

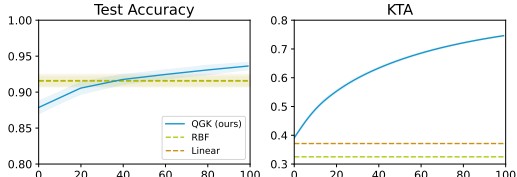

Figure 6: Scalability of test accuracy with dataset size on MNIST (10 classes, $d = 784$). QGK is compared to classical RBF and Linear Kernels using 10,000 samples.

