# OpenReview forum: "Quantum Generator Kernels"
_ICLR.cc/2026/Conference — Submitted to ICLR 2026_

### Official Review · Reviewer_H1sm · 2025-10-22

**Soundness:** 3
**Presentation:** 3
**Contribution:** 2
**Rating:** 6
**Confidence:** 4

**Summary:**

This paper introduces the Quantum Generator Kernel (QGK), a novel quantum kernel method designed to overcome the high-dimensional data-embedding bottleneck in quantum machine learning. Instead of using fixed angle encoding, they propose constructing a complete set of Hermitian generators spanning the SU(2) lie algebra. These generators are then partitined into groups, with each forming a Hermitian operator. Data dependent features are then generate the corresponding unitaries for each group. The QGK is built on top of this input embedding.

**Strengths:**

1. This work introduces a novel data encoding paradigm that moves beyond state-based methods by constructing parameterized unitary operators derived from the Lie algebra su(2^n), offering exponential increase in representational capacity; variational generator group (VGG) is proposed by structuring these unitaries through systematic combinations of algebraic generators,

2. Quantum generator kernel is defined on top of VGG with groups parametrized by a trainable linear projection; it supports standard kernel based methods like support vector machines (SVM). The parameters of the linear projection are optimized via Kernel-Target Alignment (KTA).

3. The method is benchmarked across a range of datasets with simuated noise. The results suggest that it is robust and a promising approach.

**Weaknesses:**

1. While the paper presents the Quantum Generator Kernel (QGK) as its central contribution, the more fundamental innovation appears to be the Variational Generator Group (VGG) framework for data embedding. The kernel method itself builds upon well-established concepts in quantum machine learning.

2. The VGG approach is positioned as a departure from traditional angle encoding, but it can be more precisely characterized as a generalized, structured multi-qubit angle encoding that systematically covers the su(2^n) space. This clarification would provide better context within the existing literature on quantum data embeddings and more accurately represent the methodological advancement.

3. The grouping of generators into VGGs represents a critical hyperparameter that significantly impacts the method's expressivity and efficiency. The paper would benefit from a more rigorous theoretical analysis of the grouping strategy with empirical results.

4. Another mainstream approach for encoding high-dimensional data is to use a non-linear neural net for compression followed by simpler encodings; this seems to be a more suitable approach with existing hardware in the near future. It would be good if the authors could provide theoretical analysis of the specific advanatges of the proposed encoding compared to the existing appraoch.

**Questions:**

See the weakness. I will increase my score if my concern is addressed.

**Details Of Ethics Concerns:**

No.

---

> ### Author Response · Authors · 2025-11-22
> **Response to Reviewer H1sm (1/2)**
>
> We thank the reviewer for their thoughtful, positive, and constructive feedback. We have revised the paper accordingly and summarize below how we have addressed your key concerns. We are happy to incorporate further refinements as needed.
>
> ------
>
> > While the paper presents the Quantum Generator Kernel (QGK) as its central contribution, the more fundamental innovation appears to be the Variational Generator Group (VGG) framework for data embedding.
>
> We agree that the *Variational Generator Groups* (VGGs) are central to our contribution and have updated the introduction to better reflect this. Specifically, we now emphasize:
>
> - The introduction of VGGs as a novel, structured embedding framework based on grouped Lie-algebraic generators;
> - The QGK as a generator-driven quantum kernel built atop VGGs;
> - An in-depth theoretical and empirical characterization of the expressivity, scalability, and entanglement properties of VGGs.
>
> ------
>
> > The VGG approach is positioned as a departure from traditional angle encoding, but it can be more precisely characterized as a generalized, structured multi-qubit angle encoding that systematically covers the su(2^n) space.
>
> Thank you for this perspective. We have added a clarifying discussion in the related work section to position VGGs as a principled generalization of angle encoding. Specifically, VGGs:
>
> 1. Operate over generator groups that span subspaces of $\mathfrak{su}(2^\eta)$, enabling correlated, entangling multi-qubit transformations;
> 2. Generalize standard angle encodings by parameterizing richer, data-dependent Hamiltonians instead of fixed, local rotations.

---

> > ### Author Response · Authors · 2025-11-22
> > **Response to Reviewer H1sm (2/2)**
> >
> > > The grouping of generators into VGGs represents a critical hyperparameter that significantly impacts the method's expressivity and efficiency. The paper would benefit from a more rigorous theoretical analysis of the grouping strategy with empirical results.
> >
> > We have significantly expanded both the theoretical and empirical analysis of the grouping strategy:
> >
> > - **Theory:** In a new appendix (D), we show that expressivity is invariant under any strict partitioning of the generator set. Groupings affect parameterization but not the underlying representational capacity.
> > - **Empirical:** In a new appendix (C), we evaluate various group sizes (quadratic ($\eta^2$), exponential (cf. Eq. 3), full ($4^\eta-1$)) and projection widths ($w \in [1, \eta, 2\eta]$) across datasets. We find that QGK performs robustly under these variations, with slight benefits for finer groupings and wide-stride projections.
> > - **Benchmarks:** Additional experiments in Appendix C on the *moons*, *bank*, and *MNIST* datasets confirm that these hyperparameters only mildly affect classification performance and kernel behavior.
> >
> > ------
> >
> > > Another mainstream approach for encoding high-dimensional data is to use a non-linear neural net for compression followed by simpler encodings; this seems to be a more suitable approach with existing hardware in the near future. It would be good if the authors could provide theoretical analysis of the specific advanatges of the proposed encoding compared to the existing appraoch.
> >
> > We thank the reviewer for highlighting the relevance of comparing our encoding scheme against hybrid quantum-classical models that use nonlinear neural networks for data compression — such as *Dressed Quantum Circuits* (DQC). These models are indeed compatible with current hardware and represent a pragmatic direction for near-term quantum machine learning. However, they also introduce conceptual limitations that our approach is specifically designed to avoid.
> >
> > To address this point more thoroughly, we extended the comparison to deep hybrid models in section 5 and added Appendix E (which provides a detailed theoretical comparison), clarifying:
> >
> > - Hybrid models like DQC rely on deep classical networks both before and after the quantum module. According to the universal approximation theorem, such architectures can learn arbitrary functions even when the quantum circuit performs no meaningful computation (e.g., acts as identity). This coupling makes it difficult to isolate or attribute performance gains to the quantum component.
> > - In contrast, our QGK only applies a linear affine transformation ($Wx + b$) to produce the generator coefficients that define the quantum embedding. This transformation is explicitly pre-trained to improve the Kernel Target Alignment (KTA), and it is fixed prior to classification. Consequently, the final model operates entirely within the Reproducing Kernel Hilbert Space (RKHS) induced by the quantum kernel, ensuring a clean separation between classical parameterization and quantum expressivity.
> >
> > To further quantify this separation, we added *Linear KTA*, an additional ablation which uses the same affine projection but replaces the quantum kernel with a classical linear kernel. This reveals the specific performance gains attributable to the quantum embedding. Across all benchmarks, QGK consistently outperforms Linear KTA, supporting the conclusion that the quantum kernel contributes substantially beyond the classical projection.

---

> > > ### Comment · Reviewer_H1sm · 2025-11-26
> > >
> > > Thanks for the author's detailed reply. After reading other reviewer's comments, I believe there are still places that the work needs to clarify. But I do think the new encoding strategy is interesting; it would be interesting have more theoretical analysis of the expressivity. Therefore, I would maintain the score for now.

---

> > > > ### Author Response · Authors · 2025-11-27
> > > >
> > > > Thank you for your follow-up and for recognizing the merit of our generator-based quantum kernel approach. We appreciate your engagement and remain committed to addressing any remaining concerns to further improve our work. To that end, could you kindly specify which aspects still require clarification? Any concrete suggestions would help us better focus our final revision.
> > > >
> > > > In response to your comment on theoretical analysis, we are currently incorporating a new appendix section that formalizes theoretical bounds on QGK expressivity under grouped generators. If there are specific theoretical aspects you’d like us to address, we would be grateful for your input.
> > > >
> > > > Thank you again for your time and thoughtful feedback in helping us refine this work.

---

### Official Review · Reviewer_vo9s · 2025-10-28

**Soundness:** 2
**Presentation:** 2
**Contribution:** 2
**Rating:** 4
**Confidence:** 4

**Summary:**

This paper propose a novel quantum kernel architecture. Specifically, firstly, the authors construct a set of quantum kernel generators and merge them into Variational Generator Groups (VGGs). Secondly, they introduce a linear feature extractor that is pre-trained to project high-dimensional input into a compressed generator space to improve kernel alignment. Finally, they employs Hamiltonian-driven unitaries with learnable generator weights, enabling expressive and scalable data encoding. They also provided relevant theoretical analysis.

**Strengths:**

1. The writing of this paper is good and easy to follow.

2. This paper has a certain degree of innovation, introducing a novel method for constructing quantum kernels.

3. The authors provide theoretical analysis and experimental verification of their own algorithm.

**Weaknesses:**

1. The authors are advised to conduct experiments in larger quantum systems (e.g. 20 qubits).

2. The results in Table 1 do not reflect the superiority of the quantum generator kernels (QGK). Because the dimension of moons, circles and bank are too small, it does not conform to the actual application scenario as we are in big data era. While for cifar10, the accuracy achieved by the QGK is clearly unacceptable. The reviewer suggested that the authors include additional experiments to illustrate the superiority of QGK over classical kernels.

3. The time complexity of QGK is exponentially large, and it lacks good scalability. It is difficult to run the method on quantum devices with larger qubits, which greatly reduces the practicality and impact of this method.

4. What does VQC mean on page 6, line 317?

5. For the experiment in Figure 2, the authors recommend increasing the number of circuit layers to 16 to reflect the advantages of the QGK. Because the [1] shows that the expressibility of Projected Quantum Kernel (PQK) [2] does not decrease significantly when the number of circuit layers is less than 8.

6. The reviewer is pleased to see the authors include PQK as a comparison algorithm in the entire experiment.

[1] Exponential concentration and untrainability in quantum kernel methods.

[2] Power of data in quantum machine learning

**Questions:**

Please see the weaknesses.

---

> ### Author Response · Authors · 2025-11-22
> **Response to Reviewer vo9s (1/3)**
>
> We thank the reviewer for the constructive comments and for recognizing the novelty of our proposed Quantum Generator Kernel (QGK) and its theoretical and empirical contributions. Below we address each of your concerns and outline the corresponding updates made to the paper.
>
> ------
>
> > The authors are advised to conduct experiments in larger quantum systems (e.g. 20 qubits).
>
> We fully agree that evaluating quantum kernels on larger systems is an important long-term goal. However, both current quantum hardware and classical simulators impose hard practical limits:
>
> - **Quantum hardware**: Present-day NISQ devices suffer from noise accumulation and limited connectivity. Deep circuits with many qubits (e.g., 20+) are highly error-prone, rendering results unreliable.
> - **Classical simulation limits**: The QGK constructs generator groups from the full Hermitian basis of size $4^\eta - 1$, which rapidly grows beyond tractable size. For instance, at $\eta = 10$, the full generator set already contains over one million operators, making simulation extremely memory- and compute-intensive even for a single kernel evaluation. We observe that even modest increases in $\eta$ beyond 6 require substantial resources due to the nontrivial parameter context and dense operator sums involved.
>
> To address this concern more directly, we updated the manuscript to clarify that QGK is designed not to scale in qubit count, but in **input dimension** d. Our architecture achieves **large representational capacity with few qubits**, enabling applicability to high-dimensional problems using only shallow circuits in the near term.
>
> In summary, rather than targeting unfeasibly large qubit regimes, QGK emphasizes **efficient expressivity under low-qubit, realistic hardware constraints**, and remains compatible with future fault-tolerant quantum devices where scalability may become more practical.
>
> ------
>
> > The results in Table 1 do not reflect the superiority of the quantum generator kernels (QGK). Because the dimension of moons, circles and bank are too small, it does not conform to the actual application scenario as we are in big data era. While for cifar10, the accuracy achieved by the QGK is clearly unacceptable. The reviewer suggested that the authors include additional experiments to illustrate the superiority of QGK over classical kernels.
>
> We appreciate this important remark. While small-scale benchmarks like *moons* and *circles* serve primarily as interpretable case studies to analyze kernel behavior in controlled settings, our core experiments are conducted on higher-dimensional datasets — namely *bank* (16D), *MNIST* (784D), and *CIFAR10* (3072D) — which are significantly more representative of real-world scenarios.
>
> To further substantiate the QGK’s contribution:
>
> - We added a **new MLP baseline** with a single hidden layer matching the size of the generator projection.
> - We included a **Linear KTA** baseline using the same affine projection as QGK, but with a classical linear kernel.
>
> Despite the challenging nature of CIFAR10, **QGK outperforms both the linear and neural baselines**, showing that the generator-based quantum features offer real representational benefits. While we believe this analysis demonstrates the method’s competitiveness under realistic assumptions, we would gladly include further experiments if the reviewer could recommend alternative datasets they view as more appropriate for highlighting QGK’s scalability and performance trade-offs.

---

> > ### Author Response · Authors · 2025-11-22
> > **Response to Reviewer vo9s (2/3)**
> >
> > > The time complexity of QGK is exponentially large, and it lacks good scalability.
> >
> > We appreciate this concern and are grateful for the opportunity to clarify:
> >
> > While the full Hermitian generator basis $\mathfrak{H}$ indeed contains $4^\eta - 1$ elements, the **Quantum Generator Kernel (QGK)** avoids exponential overhead by introducing a principled **grouping strategy**. This means that QGK does not use all generators directly. Instead, it aggregates them into **$g \leq 4^\eta$ variational generator groups (VGGs)**, which define the effective input dimensionality.
> >
> > Thus, we emphasize that the QGK offers **exponential expressivity per qubit**, not exponential cost. Its construction allows efficient operation on **low-qubit systems**, making it particularly suited for **near-term fault-tolerant quantum devices**. Unlike models that require many qubits to represent high-dimensional data, QGK is explicitly designed to encode large classical inputs on small-scale quantum systems.
> >
> > To bridge the gap between hardware capabilities and practical applicability:
> >
> > - We **theoretically analyze** the runtime of the QGK under classical simulation in Appendix F. Here, we show that the **QGK can be computed more efficiently than classical kernels** (e.g., RBF and linear) under sufficient generator compression, providing a favorable trade-off between expressivity and runtime.
> > - We also **clarify in Section 6** that the QGK, like all kernel methods, inherits the **$\mathcal{O}(n^2)$** complexity from pairwise kernel evaluations. We now explicitly discuss mitigation strategies, such as Nyström approximation or random feature expansion, which can be directly applied to the QGK pipeline
> >
> > >  It is difficult to run the method on quantum devices with larger qubits, which greatly reduces the practicality and impact of this method.
> >
> > We agree that scaling quantum circuits beyond 10–20 qubits remains difficult on current NISQ hardware due to noise, limited coherence, and connectivity constraints. However, this is precisely why the QGK was designed to operate **effectively in the low-qubit regime** (e.g., $\eta \in [2,5]$). Our approach offers a scalable encoding strategy for **high-dimensional classical data** on **small quantum processors**, prioritizing near-term viability and hybrid integration.
> >
> > In this sense, we believe QGK provides a practical path forward — instead of demanding more qubits, it makes **better use of fewer qubits** via structured generator parameterization and kernel optimization.
> >
> > We’ve updated the manuscript to better emphasize these points across the introduction, complexity analysis, and related work sections.
> >
> > ------
> >
> > > What does VQC mean on page 6, line 317?
> >
> > Thank you for catching this inconsistency. The term “VQC” was a leftover and should have referred to **QEK** (the variational quantum circuit baseline). We corrected this throughout the text to maintain consistency.

---

> > > ### Author Response · Authors · 2025-11-22
> > > **Response to Reviewer vo9s (3/3)**
> > >
> > > > For the experiment in Figure 2, the authors recommend increasing the number of circuit layers to 16 to reflect the advantages of the QGK. Because the [1] shows that the expressibility of Projected Quantum Kernel (PQK) [2] does not decrease significantly when the number of circuit layers is less than 8.
> > >
> > > We appreciate the reviewer’s suggestion. In Fig. 2(a), we assume a maximum of $\eta$ feasible reuploading layers, which leads to the quadratic input-scaling behaviour of the QEK shown in the figure. To ensure a fair comparison across all approaches in Fig. 2(b) and (c), the number of layers is chosen such that each model can accommodate the *maximum feasible number of input parameters* of the QGK, i.e., $4^\eta - 1$. This yields 8, 21, 64, 205, and 683 reuploading layers for 2–6 qubits, respectively.
> > >
> > >
> > >
> > > The extended evaluations in the newly added Appendix C further report the performance of the QEK under different input-scaling regimes. Specifically, we evaluate multiple group sizes (quadratic $\eta^2$, exponential as in Eq. 3, and full $4^\eta - 1$). These experiments (reported in Fig. 5) indicate that entanglement increases moderately with the number of inputs (i.e., more reuploading layers for the same qubit count). However, as reuploading grows, the overall expressibility of the resulting circuits decreases.
> > >
> > >
> > >
> > > Regarding the reviewer’s remark on the Projected Quantum Kernel (PQK), we respectfully request a clarification of the specific comparison they intend. Additional PQK-related comparisons are addressed in the subsequent response.
> > >
> > >
> > >
> > > ------
> > >
> > >
> > >
> > > > The reviewer is pleased to see the authors include PQK as a comparison algorithm in the entire experiment.
> > >
> > > Thank you for highlighting this point. We have revised the related work section and added clarifications in the main text to clearly distinguish PQK from QGK:
> > >
> > > - **PQK**, similar to HEE, first applies PCA to compress the original input to match the number of available qubits ($\eta - 1$). It then extracts one-particle reduced density matrix (1-RDM) features and constructs an RBF-like kernel based on these projected representations.
> > > - In contrast, **QGK supports direct high-dimensional encoding** without fixed classical compression, enabled by its generator-grouping scheme.
> > >
> > >
> > >
> > > We now include PQK in:
> > >
> > > - **Figure 2(a):** demonstrating that its input scalability is fundamentally limited, comparable to HEE.
> > > - **Additional empirical comparisons (Fig. 3):** confirming that PQK only performs well when label transformations are chosen to match the kernel structure (as done in [2]), and otherwise generalizes poorly.
> > >
> > >
> > >
> > > Because the PQK requires simulating measurements in three Pauli bases for each qubit to obtain the 1-RDM, it is computationally expensive. For this reason, PQK was excluded from the simulation-based noise analysis. Furthermore, its limited input scalability precludes inclusion in the entanglement and expressibility experiments: matching the input scale of QGK would require tens to thousands of qubits, which is infeasible on simulators or current hardware without aggressive classical dimensionality reduction. Such hybrid compression would obscure the intrinsic quantum embedding properties we aim to study. We now clarify these constraints explicitly in the revised text.

---

> > > > ### Comment · Reviewer_vo9s · 2025-11-28
> > > > **Official Comment by Reviewer vo9s**
> > > >
> > > > Thank you for the authors’ reply. Most of my concerns have been dispelled. However, I still have a question I'd like to discuss with the authors. A positive response would be greatly appreciated, as it would improve this paper's score. Does QGK still exhibit advantages as the number of qubits increases (balancing overhead and performance)? Or could you provide a theoretical analysis of the appropriate range for QGK usage within a certain number of qubits, and beyond that, should we design new quantum kernels or use other types of quantum kernels? Because eventually we will move towards an era with a large number of qubits. And embracing the quantum revolution.

---

> > > > > ### Comment · Reviewer_H1sm · 2025-11-28
> > > > >
> > > > > I would agree with the reviewer vo9s. For me, a threotical analysis of the expressive power induced by the multi-qubit encoding gate could be very interesting.

---

> > > > > > ### Author Response · Authors · 2025-11-30
> > > > > >
> > > > > > Thank you for acknowledging our previous replies, engaging in the discussion, and raising this important question.
> > > > > >
> > > > > > To answer directly: **yes, the advantageous scaling of the QGK persists for arbitrary numbers of qubits**, since the available generator space grows as $4^\eta-1$, allowing increasingly rich data embeddings without any intrinsic theoretical limitation.
> > > > > >
> > > > > > Our exponential grouping strategy aggregates these generators into balanced groups, which preserves expressivity while keeping the parameter overhead manageable. To support this formally, we have added a new theoretical analysis in the revised appendices (updated Appendices D and E). These additions provide explicit *upper and lower bounds* on the kernel expressivity of the QGK as $\eta$ increases. In summary, they show that:
> > > > > >
> > > > > > 1. **The lower expressivity bound decays exponentially with the number of qubits**, ensuring that the QGK can become increasingly expressive as system size grows — provided grouping remains balanced (which is ensured under our exponential scheme).
> > > > > > 2. **The upper bound is controlled by the group size**, which remains controlled under exponential grouping. This guarantees stable kernel behaviour even for large qubit counts and highlights that QGK continues to be advantageous as systems scale.
> > > > > > 3. **The affine preprocessing layer further improves expressivity** by reweighting generator directions in a way that reduces anisotropy. Importantly, this tightens the upper bound without adding any classical nonlinear expressivity.
> > > > > >
> > > > > > Overall, the new theoretical results confirm that **QGK maintains high expressivity, stability, and parameter efficiency for increasing numbers of qubits**, and that exponential grouping is particularly well-suited for balancing overhead and performance in large-scale settings. This makes the QGK a **highly suitable kernel approach for future large-scale fault tolerant quantum systems**, where increasing qubit counts can be fully leveraged without sacrificing stability or efficiency.
> > > > > >
> > > > > > We hope these additions fully address your request for a deeper theoretical analysis of QGK behaviour in the multi-qubit regime.

---

### Official Review · Reviewer_H96o · 2025-10-31

**Soundness:** 2
**Presentation:** 2
**Contribution:** 3
**Rating:** 4
**Confidence:** 4

**Summary:**

This paper proposes Quantum Generator Kernels (QGK), a new approach to quantum kernel methods aimed at efficiently embedding high-dimensional data into quantum feature spaces. The key idea is to use Variational Generator Groups (VGGs), which are sets of Lie algebra generators combined into parameterized Hermitian operators, to build expressive quantum feature maps. The authors use a classical linear feature extractor to compress the input into generator coefficients, which then weight the quantum generators to produce a quantum state. The kernel value is computed via state fidelity. This approach is highly configurable and aims to span the full Hilbert space expressivity. The authors provide theoretical analysis and empirical validation on synthetic and real-world (MNIST/CIFAR-10) datasets. QGK consistently outperforms classical and other quantum kernels, notably even under simulated hardware noise.

**Strengths:**

QGK directly tackles the practical and important issue of embedding high-dimensional data into quantum states, which is a major bottleneck for QML.
The use of Variational Generator Groups (VGGs) is a fresh and powerful idea, allowing for a learnable feature map rather than a fixed one. This is a commendable cross-disciplinary innovation.
The fact that QGK’s advantage persists under noise simulations is a major strength. This demonstrates robustness and suggests viability for NISQ devices, which is a significant step beyond many "ideal simulation" papers.

**Weaknesses:**

The method relies on a classical linear compression stage before the quantum kernel. This blurs the line of quantum advantage. It's unclear how much of the performance gain comes from the trained classical pre-processing versus the quantum kernel itself. The paper needs a stronger ablation study to disentangle these two contributions.
Experiments on MNIST/CIFAR-10 use small subsets (n=1000). This is insufficient to demonstrate scalability. Kernel methods are notoriously difficult to scale with the number of training samples (N), often requiring O(N^2) or O(N^3) complexity, and this critical weakness is not addressed at all.
Sensitivity to design choices (number of generator groups $g$, compression ratio $\gamma$) is not deeply analyzed. This makes it hard for a practitioner to know how to tune this model.
The comparison is mostly versus kernel baselines, not small neural nets.

**Questions:**

I suggest the authors clarify, in a sentence or two, how the classical compression is obtained and whether it is fixed. Furthermore, a more detailed ablation study is needed to disentangle the effects of this compression stage from the QGK's contribution.
Please comment briefly on whether you expect QGK to scale to larger training sets (e.g., full MNIST). The computational complexity with respect to $N$ (number of samples) seems to be the elephant in the room.
A short remark on how performance changes when you vary the number of generator groups would help readers judge tuning effort.
It would help to summarize (even approximately) the qubit count and circuit depth for your best MNIST/CIFAR setting.
A one-line comparison to a tiny MLP/CNN on the same 1k-sample split would help contextualize absolute accuracy.

---

> ### Author Response · Authors · 2025-11-22
> **Response to Reviewer H96o (1/3)**
>
> Thank you for your constructive and detailed feedback. We appreciate your recognition of the practical relevance of QGK, its robustness under simulated noise, and the novelty of our generator-based approach. Below we address your key concerns and summarize the substantial additions made to the revised manuscript. We are happy to clarify any remaining questions and hope the updates presented below sufficiently address your concerns to warrant a positive reassessment of your score.
>
> ---
>
> ## Baselines and Ablations
>
>
> > The method relies on a classical linear compression stage before the quantum kernel. This blurs the line of quantum advantage. It's unclear how much of the performance gain comes from the trained classical pre-processing versus the quantum kernel itself. The paper needs a stronger ablation study to disentangle these two contributions.
>
> We agree that disentangling classical and quantum contributions is essential, especially given the prevalence of heavy classical preprocessing in hybrid QML architectures. Importantly, the QGK uses **only a single linear layer** (Wx + b) implemented via a torch.nn.Linear. This layer only **parameterizes generator weights** and **does not introduce any nonlinear expressivity**. As such, it cannot act as a universal approximator, distinguishing QGK from hybrid models that combine deep classical and quantum layers.
>
> To address this issue more thoroughly, we extended Section 5 to explicitly compare the QGK to hybrid architectures like the *Dressed Quantum Circuit* (DQC), which incorporate nonlinear classical networks both before and after quantum processing. As detailed in the new Appendix E, according to the universal function approximation theorem, the classical components of such models alone can represent any continuous function — even when the quantum module acts trivially. This makes performance attribution inherently ambiguous.
>
>
> > I suggest the authors clarify, in a sentence or two, how the classical compression is obtained and whether it is fixed.
>
> This has been clarified in the main text (Section 4) with further details provided in Appendix E. Specifically:
>
> QGK ensures a clean separation between classical and quantum components:
>
> - The classical compression is a **linear affine transformation** $Wx + b$, no classical post-processing or deep nonlinear layers are involved.
> - The affine projection is trained **only** to maximize **Kernel Target Alignment (KTA)** but remains **fixed** during classification via the SVM.
> - The resulting model performs classification **entirely within the RKHS** defined by the induced quantum kernel $\mathcal{K}_{\phi}$.
>
>
> > Furthermore, a more detailed ablation study is needed to disentangle the effects of this compression stage from the QGK’s contribution.
>
> Thank you for this suggestion. We expanded our ablation study to include **Linear KTA**, which uses the same classical projection as the QGK but replaces the quantum kernel with a classical linear kernel. This baseline, together with the previously included **HEE-Linear** (applying the same projection to a hardware-efficient encoding) and **QGK-Static** (no projection training), isolates and quantifies the quantum kernel’s contribution.
>
> The updated results consistently show that QGK outperforms all ablations, confirming that **the quantum component adds meaningful representational capacity** beyond what is achievable by the classical projection alone.
>
>
>
> > The comparison is mostly versus kernel baselines, not small neural nets. [...] A one-line comparison to a tiny MLP/CNN on the same 1k-sample split would help contextualize absolute accuracy.
>
> Per your suggestion, we added an MLP baseline, a compact neural network with a single hidden layer of size $g$ (equal to the number of generator groups), trained with cross-entropy loss. QGK surpasses this MLP across all tasks, including CIFAR‑10.
>
>
>
> The following table summarizes the final accuracies of the addtional ablations and baselines comared to the QGK across all benchmarks:
>
> | Dataset | QGK (ours) | Linear KTA | MLP $(d \to g \to c)$ |
> |-|:-:|:-:|:-:|
> | moons (d=2, g=15, c=2)| **0.96 ± 0.04** | 0.86 ± 0.05 | 0.87 ± 0.06 |
> | circles (d=2, g=15, c=2) | **0.68 ± 06** | 0.44 ± 0.09 | 0.64 ± 0.11 |
> | bank (d=16, g=15, c=2) | **0.86 ± 0.07** | 0.76 ± 0.06 | 0.78 ± 0.09  |
> | MNIST (d=784, g=93, c=10)| **0.88 ± 0.03** | 0.87 ± 0.03 | 0.87 ± 0.04 |
> | CIFAR10 (d=3072, g=93, c=10)| **0.38 ± 0.05** | 0.32 ± 0.03 | 0.29 ± 0.06 |
>
>
> These results demonstrate that QGK consistently outperforms both a purely classical projection + kernel and a compact neural baseline, especially on higher-dimensional tasks. This strengthens evidence that the learned projection does *not* overshadow the quantum component.

---

> > ### Author Response · Authors · 2025-11-22
> > **Response to Reviewer H96o (2/3)**
> >
> > ## Input Scaling & Kernel Complexity
> >
> > > Experiments on MNIST/CIFAR-10 use small subsets (n = 1000). This is insufficient to demonstrate scalability. Kernel methods are notoriously difficult to scale with the number of training samples (N), often requiring O(N²) or O(N³) complexity, and this critical weakness is not addressed at all.
> >
> > We appreciate the reviewer’s concern regarding scalability and agree that computational complexity with respect to the number of training samples n remains a core challenge for kernel methods. In our work, the proposed QGK specifically targets scalability with respect to **input dimensionality** d, not n, by providing a flexible and efficient embedding mechanism for high-dimensional data using low-qubit counts.
> >
> > To address potential misunderstandings and improve clarity, we have revised the paper as follows:
> >
> > - **Clarified our contribution focus** in the introduction: We now explicitly state that QGK is designed to support **efficient high-dimensional input encoding** under qubit constraints.
> > - **Extended the computational complexity discussion** in the empirical analysis (Section 6) to highlight that the QGK, like all kernel methods (including QEK, HEE, RBF, and Linear), inherits the standard $\mathcal{O}(n^2)$ complexity from pairwise kernel evaluations. We also outline potential strategies for mitigating this cost, such as **Nyström approximation** or **random feature expansion**, which are compatible with the QGK.
> >
> >
> > > Please comment briefly on whether you expect QGK to scale to larger training sets (e.g., full MNIST).
> >
> > We expect the QGK to scale well to larger datasets, as its design enables efficient encoding of high-dimensional inputs using limited-qubit circuits. To provide initial empirical support, we added results on **10,000 samples** from the MNIST dataset in **Appendix G**, comparing QGK to classical baselines. This evaluation shows:
> >
> > - **QGK reaches 94% ± 1% test accuracy**, outperforming RBF (**92% ± 1%**) and Linear kernels (**91% ± 1%**).
> > - Performance scales smoothly with increasing n, while maintaining practical runtime and tight confidence intervals.
> >
> >
> > These results suggest that the QGK retains its performance benefits at larger scales, comparable to traditional kernel methods in terms of sample complexity. We anticipate even broader applicability when combined with standard kernel approximations (e.g., Nyström), which we plan to explore in future work.

---

> > > ### Author Response · Authors · 2025-11-22
> > > **Response to Reviewer H96o (3/3)**
> > >
> > > ## **Hyperparameter Sensitivity**
> > >
> > > > Sensitivity to design choices (number of generator groups, compression ratio) is not deeply analyzed. This makes it hard for a practitioner to know how to tune this model.
> > >
> > > Thank you for highlighting this important point. In response, we have significantly extended both the theoretical and empirical analysis of the generator grouping and projection strategies to support informed model tuning.
> > >
> > > In the new **Appendix C**, we now analyze performance, expressivity, entanglement, and circuit depth under different configurations:
> > >
> > > - **Group Scaling Strategies:** We compare quadratic scaling ($g = \eta^2$), exponential scaling ($g$ as per Eq. 3, default), and full-generator grouping ($g = 4^\eta - 1$) showing the QGK's kernel properties are robust accross configurations.
> > > - **Projection Widths:** We test wide ($w=1$), medium ($w=\eta$), and narrow ($w=2\eta$) strides, showing a **wide stride ($w=1$)** consistently yields stable and expressive kernels.
> > > - **Larger groups** (e.g., quadratic scaling) may be slightly more sensitive, while exponential and full-generator groupings offer consistently stable expressivity across projection widths.
> > > - **Benchmarks:** Experiments across *moons*, *bank*, and *MNIST* demonstrate that group size and stride cause only **modest performance variation**. On smaller tasks like \texttt{moons}, the model is virtually insensitive to grouping. For high-dimensional data, exponential scaling offers the best trade-off between stability and performance.
> > > - **Circuit depth remains unaffected** by the grouping strategy, as the total number of generators remains unchanged, only their structural organization is altered.
> > >
> > >
> > > In the new **Appendix D**, we provide a formal theoretical analysis of generator groupings. Key insights include:
> > >
> > > - **Any strict partition** of the Hermitian basis preserves kernel expressivity.
> > > - **Linear independence** among grouped operators is helpful but not strictly required.
> > >
> > >
> > >
> > > Together, these results clarify that the grouping strategy affects the *parameterization*, not the *representational power* of the QGK.
> > >
> > >
> > >
> > >
> > >
> > > > *“A short remark on how performance changes when you vary the number of generator groups would help readers judge tuning effort.”*
> > >
> > > We extended Sections 3 and 6 by the key insights form the additional results in Appendix C and D summarized above. In summary: performance remains stable across group counts, with slight improvements for exponential or full-group scaling. In practice, exponential scaling offers the best balance between stability, expressivity, and simulation cost.
> > >
> > >
> > >
> > >
> > >
> > > >  It would help to summarize (even approximately) the qubit count and circuit depth for your best MNIST/CIFAR setting.
> > >
> > >
> > >
> > > Thank you for the helpful suggestion. Table 1 of our submission reports the compiled circuit depth for the key evaluated approaches (shown in parentheses after each accuracy value). In response to your feedback, we have extended this table to explicitly include the number of qubits used for each benchmark. Additionally, we now provide a detailed breakdown of the qubit count, level-1- and level-2-optimized circuit depths for all evaluated configurations across benchmarks in Table 6 (Appendix G).

---

### Author Response · Authors · 2025-11-22
**Rebuttal Revision Summary**

We sincerely thank all reviewers for their constructive and insightful feedback. We have carefully revised the manuscript to address the raised concerns and incorporate the suggested improvements. Below we summarize the changes; additions in the updated paper are highlighted in **blue**:

- **Introduction:** Revised the contribution list to more clearly position and contextualize our work. *(Reviewers H1sm, H96o, vo9s)*
- **Theoretical and empirical elaborations:** Added extended theoretical analysis (Section 3 and new Appendix D) and additional empirical insights (Section 6 and new Appendix C) on the effect of different group sizes and projection widths on the kernel’s theoretical properties. Appendix C provides extensive empirical comparisons across hyperparameters, while Appendix D characterizes the theoretical impact of grouping. *(Reviewers H1sm, H96o)*
- **Scalability and sample complexity:** Clarified the scaling behavior with respect to the number of training samples and proposed mitigation strategies (Section 6). Appendix G has been expanded with additional experiments demonstrating QGK performance under increased training-set sizes. *(Reviewer H96o)*
- **Pre-processing details:** Extended Section 4 with a more detailed description of the applied pre-processing pipeline. Introduced a new Appendix E providing a theoretical characterization of the effect of the affine transformation on the resulting kernel. *(Reviewers H1sm, H96o)*
- **Comparative theoretical framing:** Expanded Section 5 by reframing QGK as a generalized multi-qubit angle-encoding method and adding theoretical comparisons to the *Dressed Quantum Circuit* (DQC) and the *Projected Quantum Kernel* (PQK). *(Reviewers H1sm, vo9s)*
- **Extended baselines:** Augmented the empirical evaluation (Section 6 and Appendix G) with additional baselines, including the Projected Quantum Kernel, a lightweight multi-layer perceptron, and a classical linear kernel pre-trained using Kernel Target Alignment (KTA). *(Reviewers H96o, vo9s)*

---

### Author Response · Authors · 2025-12-02
**Overview for the New Area Chair: Key Revisions and Review Status**

Dear Area Chair, thank you very much for stepping in to handle our submission under the unusual circumstances caused by the recent incident. We greatly appreciate your engagement in thoroughly evaluating the paper given that reviews and scores reverted to their pre-discussion state, and reviewers were no longer able to update their assessments during the final stage of the rebuttal.

We would also like to sincerely thank all reviewers once again, who, despite the unforeseen situation, provided thoughtful, detailed, and constructive feedback that substantially strengthened our work. Importantly, two reviewers explicitly indicated that several of their concerns had been addressed and were considering increasing their scores, but were unable to submit final updates due to the forced discussion freeze. We have incorporated those remaining clarifications and additions into the final revision for your consideration.

Below we provide a summary of the revisions made during the rebuttal (all additions highlighted in **blue** in the manuscript):

- **Clarified positioning and contributions**: We reworked the introduction and contribution list to better distinguish the Variational Generator Group (VGG) framework from the Quantum Generator Kernel (QGK), and to clearly position VGG as a generalized, structured multi-qubit angle encoding, situating our work more explicitly within existing QML embedding methods.
- **Expanded theoretical analysis of grouping and expressivity**: We added new appendices (especially C and D) that analyze the effect of generator groupings and projection widths, both theoretically and empirically. These results show that strict partitions of the generator set preserve expressivity, characterize how different grouping strategies impact kernel behavior, and provide new bounds on expressivity as the number of qubits grows, directly addressing the follow-up questions by vo9s and H1sm about multi-qubit behaviour and long-term scalability.
- **Clearer treatment of classical preprocessing vs quantum contribution**: We clarified that the classical stage is a single affine projection trained via Kernel Target Alignment (KTA) and fixed during SVM training, with no nonlinear classical layers involved. To disentangle contributions, we added ablations using the same projection with a purely classical linear kernel (Linear KTA), alongside the existing HEE-Linear and QGK-Static variants. Across all benchmarks, QGK consistently outperforms these, indicating a genuine quantum contribution. In addition, the new Appendix E provides a theoretical analysis showing how the affine preprocessing reweights generator directions, reducing anisotropy and tightening the upper expressivity bound without adding classical nonlinearity.
- **Broadened empirical evaluation**: As suggested, we added a compact MLP baseline and an explicit PQK baseline, both of which QGK consistently outperforms, clarifying its advantage over classical and alternative quantum kernels. Appendix C further includes additional hyperparameter sweeps over group sizes and projection widths, showing that the QGK remains robust across configurations, with exponential grouping offering the most stable balance of expressivity and computational overhead.
- **Complexity and scalability**: We clarified that the QGK is designed to encode high-dimensional inputs using few qubits, making it suitable for near-term small-scale fault-tolerant hardware, robust on current noisy devices (as demonstrated in our noise simulations), and efficiently executable fully classically when hardware is limited — thereby bridging the gap to future devices. While the QGK scales smoothly with increasing qubit numbers, it does not rely on large qubit counts to be effective. We further acknowledge that the QGK inherits the standard sample-scaling of kernel methods and outline compatible mitigation strategies (Nyström approximation, random features). Importantly, the generator-grouping strategy and affine preprocessing avoid exponential overhead in practice, enabling applicability today while remaining scalable for future larger-qubit regimes. Finally, Appendix G now includes preliminary results on a larger MNIST subset, providing additional evidence for scalability in the number of training samples.

- **Minor corrections and clarifications:** We fixed inconsistent terminology, expanded the description of the preprocessing pipeline, and summarized qubit counts and circuit depths for all main configurations to improve reproducibility and readability.

We hope this summary assists your assessment of the submission under the special conditions of the reassignment.

---

### Meta-Review · Area_Chair_PHyZ · 2025-12-31

**Summary:**

This paper proposed Quantum Generator Kernels (QGKs), a generator-based approach to quantum kernels. In the initial version, the reviewers raised questions and concerns about:
- Comparison to classical compression methods
- Novelty over existing quantum machine learning methods, especially that the proposed Variational Generator Groups (VGGs) are more like direct generalization of variational quantum algorithms
- Lack of theory justifications

During the rebuttal, the authors made notable clarifications and revisions to the paper, including:
- Introduction: More detailed comparison to earlier works.
- Theoretical and empirical elaborations: Added extended theoretical analysis (Section 3 and new Appendix D) and additional empirical insights (Section 6 and new Appendix C) on the effect of different group sizes and projection widths on the kernel’s theoretical properties.
- Scalability and sample complexity: Clarified the scaling behavior with respect to the number of training samples and proposed mitigation strategies (Section 6).
- Pre-processing details: Extended Section 4 with a more detailed description of the applied pre-processing pipeline.
- Comparative theoretical framing: Expanded Section 5 by reframing QGK as a generalized multi-qubit angle-encoding method and adding theoretical comparisons to the Dressed Quantum Circuit (DQC) and the Projected Quantum Kernel (PQK).
- Extended baselines: Augmented the empirical evaluation (Section 6 and Appendix G) with additional baselines, including the Projected Quantum Kernel, a lightweight multi-layer perceptron, and a classical linear kernel pre-trained using Kernel Target Alignment (KTA).

Given the borderline scores and the scenario of the conference this year, the AC carefully checked the paper. It is noted that although the authors made decent efforts on revising the paper, the following concerns remain:
- The scalability of the results are unclear: Regarding the reivewer vo9s's question that whether the experiments can reach 20 qubits, the authors pointed out in the rebuttal that this is difficult both for quantum hardware due to noise accumulation and limited connectivity, and also for classical simulation due to $4^{\eta}$ grows too fast with $\eta$. However, it is noted that for many quantum experiments in practice, such as Google's quantum supremacy experiments in 2019 and many more recent experiments by Google/IBM/Quantinuum and others, many interesting experiments have been conducted on say 50+ qubits. If the studied algorithm cannot reach 20 qubits, it probably means that it has less impact at least in the near term.
- Theory: Speaking of long term impacts, the paper has theory results, but to AC's opinion it's more on justification of specific perspectives such as grouping and expressivity, but not clear whether the proposed method has theoretical advantage as a whole.

Considering these as welll as the borderline scores (two negative scores of 4, no very positive score of at least 8), the decision is rejection at ICLR 2026.

**Reviewer Concerns:**

As far as I see, the authors did a decent job in addressing reviewer concerns. I didn't see outstanding issues.

**Reviewer Scores:**

This is hard to predict for this paper - I think the authors did a decent job, but there are also essential issues with the paper as mentioned in the meta-review. Maybe the scores will marginally increase, but it won't overturn the decision by the AC.

---

### Decision · Program_Chairs · 2026-01-26

Reject